# Shaping of inner ear sensory organs through antagonistic interactions between Notch signalling and Lmx1a

Zoe F Mann, Héctor Gálvez[†], David Pedreno[†], Ziqi Chen, Elena Chrysostomou, Magdalena Żak, Miso Kang, Elachumee Canden, Nicolas Daudet*

The Ear Institute, University College London, London, United Kingdom

**Abstract** The mechanisms of formation of the distinct sensory organs of the inner ear and the non-sensory domains that separate them are still unclear. Here, we show that several sensory patches arise by progressive segregation from a common prosensory domain in the embryonic chicken and mouse otocyst. This process is regulated by mutually antagonistic signals: Notch signalling and Lmx1a. Notch-mediated lateral induction promotes prosensory fate. Some of the early Notch-active cells, however, are normally diverted from this fate and increasing lateral induction produces misshapen or fused sensory organs in the chick. Conversely Lmx1a (or cLmx1b in the chick) allows sensory organ segregation by antagonizing lateral induction and promoting commitment to the non-sensory fate. Our findings highlight the dynamic nature of sensory patch formation and the labile character of the sensory-competent progenitors, which could have facilitated the emergence of new inner ear organs and their functional diversification in the course of evolution.

DOI: https://doi.org/10.7554/eLife.33323.001

*For correspondence: n.daudet@ucl.ac.uk

[†]These authors contributed equally to this work

Competing interests: The authors declare that no competing interests exist.

## Introduction

The morphogenesis of tissues with a complex three-dimensional architecture and a variety of specialized cell types requires the coordinated action of mechanisms that specify cell identity and position. The vertebrate inner ear is a formidable model system in which to investigate these processes (*Groves and Fekete, 2012*; *Alsina and Whitfield, 2017*). It hosts multiple specialised sensory organs responsible for hearing in the cochlea, and the perception of head movements and position in the vestibular system. Each sensory organ contains a sensory patch, composed of a regular mosaic of mechanosensory 'hair' cells, and their supporting cells. In between these sensory patches are non-sensory domains, forming a series of fluid-filled canals and constrictions essential for inner ear function.

The morphology of the inner ear, as well as the number, type, position and shape of sensory patches that it hosts, vary greatly across and within distinct classes of vertebrates (*Baird, 1974*; *Schulz-Mirbach and Ladich, 2016*; *Manley and Clack, 2004*). The inner ear of ancestral vertebrates was probably similar to that of hagfish (*Jørgensen et al., 1998*) and lampreys (*Lowenstein et al., 1968*; *Maklad et al., 2014*), with a relatively small number of sensory patches (three in hagfish) and a rudimentary vestibular system. In the course of evolution, a gradual increase in the number of sensory patches (up to nine in some amphibians), possibly due to the duplication and modification of pre-existing structures, has led to the acquisition of new inner ear functionalities – in particular sound detection in terrestrial vertebrates (*Manley and Clack, 2004*; *Fritzsch et al., 2002*; *Manley, 2012*). For example, the vestibular system of birds and mammals comprises three patches (the cristae) connected to the semi-circular canals, two large patches (the maculae) in the saccule and the utricle, and the smaller macula neglecta. In mammals, the auditory cochlea forms a coiled tube that contains

one sensory epithelium called the organ of Corti. In birds, the ventral portion of the inner ear is formed by an elongated (uncoiled) recess containing an auditory epithelium, the basilar papilla, as well as one vestibular-like sensory patch at its distal end, the lagena. The monotremes (the platypus and echidnas), remarkably, exhibit intermediate characteristics: an elongated cochlear duct, containing an organ of Corti-like epithelium and a distal lagena, suggesting that the lagena was present in ancestral mammals, but lost in placental and marsupial mammals (*Schultz et al., 2017*; *Ladhams and Pickles, 1996*). The genetic changes that have led to such diversification of sensory organs during vertebrate evolution are still elusive, but are necessarily connected to the factors guiding their formation during embryonic development. Interestingly, it has been proposed that developing sensory patches could form progressively by segregation from a common 'pan-sensory' domain, a process that would recapitulate the evolutionary history of the inner ear (*Fritzsch et al., 2002*; *Knowlton, 1967*). Whilst this hypothesis has received some experimental support from anatomical studies, in particular in amphibians (*Fritzsch and Wake, 1988*; *Norris, 1892*), the precise molecular mechanisms regulating sensory patch number and position, or their segregation remain unclear.

All sensory patches are thought to derive from so-called 'prosensory' domains, found in the anterior and posterior domains of the otocyst, the epithelial vesicle giving rise to the inner ear. The prosensory domains are characterised by expression of the Notch ligand Jagged1 (Jag1, also known as Serrate1 in the chick) and of the transcription factor Sox2, which are both required for sensory domain formation (*Kiernan et al., 2001*; *Kiernan et al., 2006*; *Kiernan et al., 2005*; *Brooker et al., 2006*; *Neves et al., 2007*; *Cole et al., 2000*). Notch activity, induced by Jag1, promotes Sox2 expression and commitment to the sensory fate (*Pan et al., 2010*; *Daudet and Lewis, 2005*; *Hartman et al., 2010*; *Neves et al., 2011*). In addition, Jag1 is positively regulated by Notch, through a process known as 'lateral induction' (*Eddison et al., 2000*; *Lewis, 1998*). Besides maintaining Notch activity within the prosensory domains, lateral induction could in theory lead to the propagation of Notch activity across interacting cells and play an important role in regulating the size, number and shape of inner ear sensory patches. However, uncharacterized signals must also act to counteract lateral induction and prevent excessive expansion of the developing sensory domains.

In this study, we set out to investigate the role of lateral induction during early sensory patch formation. We show that several sensory organs form by segregation from a common prosensory domain in the early developing chick otocyst, and found that some of the early Notch-active (and presumably sensory-competent) cells can change character and become non-sensory cells in the mature inner ear. We also show that a gain of lateral induction disrupts the normal positioning of sensory patch boundaries, and in extreme cases, induces fusion of adjacent sensory patches. This suggests that a dampening of lateral induction is required for the normal segregation of inner ear sensory patches. We next demonstrate that the LIM-homeodomain transcription factor Lmx1a (cLmx1b in the chicken), expressed in non-sensory domains, antagonizes lateral induction and promotes the non-sensory fate in a context-dependent manner. The expression of cLmx1b is negatively regulated by Notch activity, and analysis of an Lmx1a- GFP knock-in mouse model suggests that some sensory cells at the lateral border of sensory patches are derived from Lmx1a-expressing cells.

Altogether, these results suggest that the embryonic segregation of sensory patches and the positioning of their boundaries occur progressively during development and is regulated by the balance of mutually antagonistic signals promoting (e.g. lateral induction, Sox2) and antagonizing (Lmx1a, among others) commitment to the sensory fate. The labile character of the early sensory-competent progenitors could have facilitated the emergence of a wide variety of sensory organ types and morphologies in the course of vertebrate evolution.

## Results

### The anterior and lateral cristae form by segregation from a large prosensory domain during development of the chick inner ear

The Notch ligand Jag1 is one of the earliest markers of the prosensory domains of the inner ear. In the chicken otocyst at embryonic day (E) 2.5 (*Figure 1A*), Jag1 immunostaining is elevated in the neurosensory competent anterior domain and in the posterior domain. Over the subsequent days of development, Jag1 becomes elevated in defined prosensory domains, which increase in size and

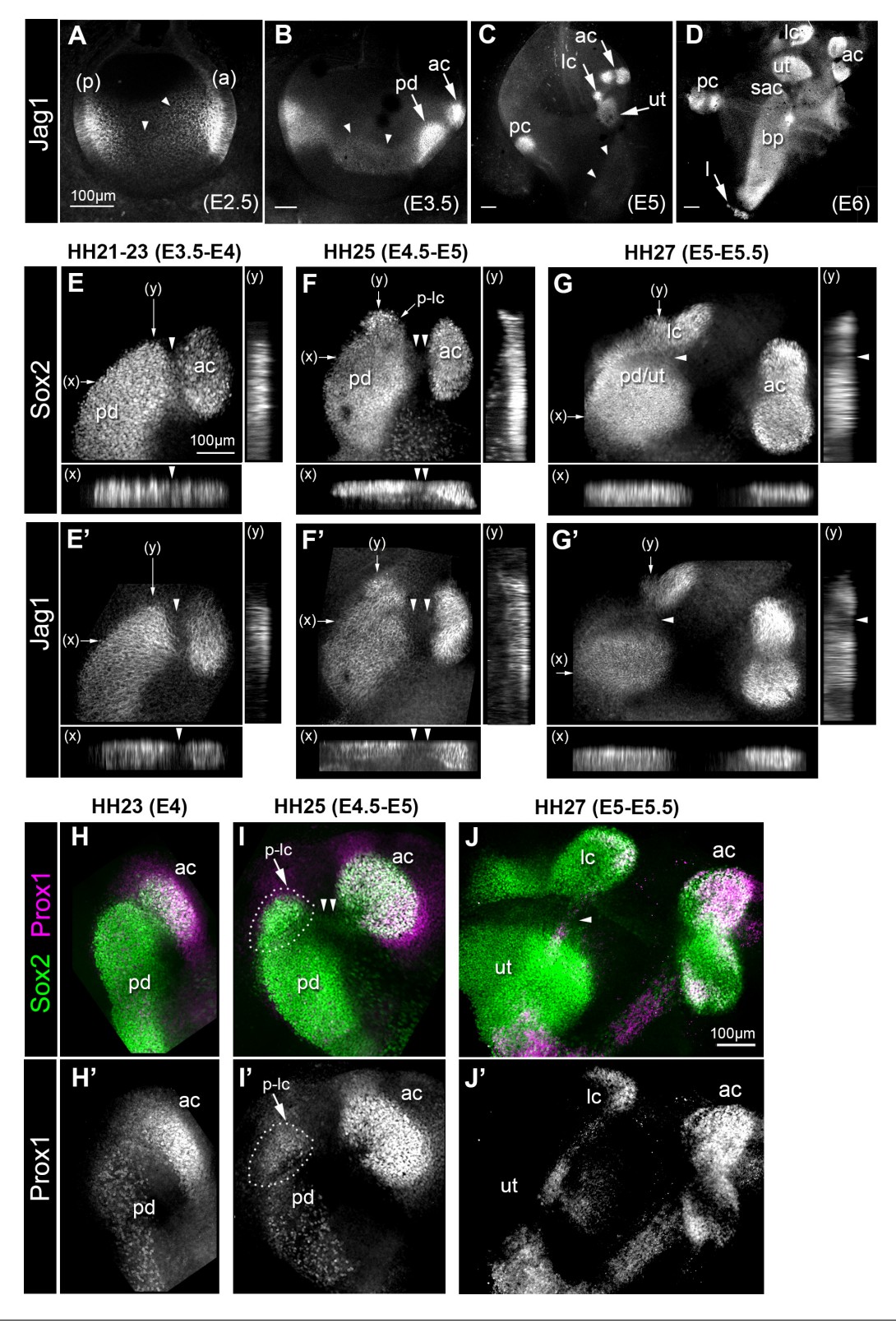

**Figure 1.** Expression of the Notch ligand Jag1,Sox2 and Prox1 during the formation and segregation of the chick inner ear sensory patches. (A–D) Jag1 expression in the developing chick inner ear (whole-mount preparations). At E2.5 (A), two patches of high expression are present in the anterior (a) and posterior (p) poles of the otocyst, but lower levels of expression are also detected in the medial region along the antero-posterior axis (arrowheads). At E3.5 (B), two patches of high Jag1 expression are now present in the anterior domain (arrows), corresponding to the anterior crista (ac) and a larger

*Figure 1 continued on next page*

*Figure 1 continued*

prosensory domain (pd); the medial domain of lower expression remains clearly visible (arrowheads). At E5 (**C**), strong Jag1 staining is observed in the posterior (pc), anterior and lateral (lc) cristae and the utricle (ut); another fainter domain of expression extends ventrally within the developing cochlear duct (arrowheads). At E6 (**D**), the utricle, the three cristae, and the lagena (l) are well defined. Jag1 expression still forms a continuous domain of expression between the saccule (sac) and the basilar papilla (bp). (**E–G'**) Whole-mount surface views and transverse projections (along the x and y directions) of the anterior prosensory domains at different developmental stages, immunostained for Sox2 and Jag1 expression. At HH21-23 (**E–E'**), the prospective anterior crista becomes apparent at the edge of a large prosensory domain (pd); a group of Sox2-positive and Jag1-negative cells (arrowheads) is located at the interface of the two prosensory patches. At HH25 (**F–F'**), the distance separating the anterior crista from the large prosensory domain increases (double arrowheads); a distinct cluster of Sox2-positive cells becomes visible at the upper edge of the large prosensory domain, which corresponds to the location of the prospective lateral crista (**p–lc**). At HH27 (**G–G'**), the lateral crista starts to segregate from the prospective utricle (pd/ut); the interface domain is Sox2-positive, but exhibits a marked down-regulation of Jag1 expression (arrowheads). At HH23 (**H–H'**), Prox1 is strongly expressed in the prospective anterior crista (ac), which is still connected to the large prosensory domain (pd). At HH25 (**I–I'**), strong Prox1 immunostaining is present in the anterior crista; a new, denser cluster of Prox1-positive nuclei (dotted outline) is also present at the superior edge of the large prosensory domain, presumably corresponding to the prospective lateral crista (**p–lc**). Note that the cells located in between the anterior crista and the large prosensory domain still retain faint Sox2 expression (arrowheads in I). At HH27 (**J–J'**), the lateral crista (lc) segregates; note that only one of its halves contains Prox1-expressing cells at this stage. Scale bar for all panels = 100 µm.

DOI: https://doi.org/10.7554/eLife.33323.002

The following source data and figure supplements are available for figure 1:

**Figure supplement 1.** Quantification of anterior and lateral cristae segregation in the embryonic chick inner ear.
DOI: https://doi.org/10.7554/eLife.33323.003
**Figure supplement 1—source data 1.** Measurements of the surface area of the anterior and lateral cristae and the distance separating them from the anterior prosensory domain as a function of the developmental stage (Hamburger and Hamilton stages) for each sample analyzed.
DOI: https://doi.org/10.7554/eLife.33323.004

number as the inner ear matures (*Figure 1B–D*). However, as previously described (*Cole et al., 2000*), low levels of Jag1 expression are transiently detected in a broad ventro-medial domain that extends along the antero-posterior domain of the otocyst between E2.5 and E3.5 (arrowheads in *Figure 1A–B*). The proximity of the anterior and lateral crista to the prospective utricle at early stages of development (*Figure 1B–C*) prompted us to examine in greater detail, using whole-mount preparations, the relationship between Sox2 and Jag1 expression during their formation. In samples examined between HH21-23 (E3.5-E4), a distinct cluster of Sox2/Jag1-expressing cells (the presumptive anterior crista, based on its location and evolution at later stages) was located next to a larger prosensory domain (*Figure 1E–E'*). The junctional domain between the two adjacent patches was slightly constricted, and contained Sox2-positive (Sox2+) but Jag1-negative (Jag1-) cells (arrowheads in *Figure 1E–E'*). At HH25 (E4.5 -E5), there was a marked down-regulation in Sox2 expression between the anterior crista and the prosensory domain (double arrowheads in *Figure 1F–F'*), and an increase in the distance separating these patches. At the same time, a distinct cluster of Sox2+ cells (the presumptive lateral crista) became visible at the superior edge of the prosensory domain (arrow in *Figure 1F*). At HH27 (E5-E5.5), a decrease in Sox2 expression was seen at the interface of the lateral crista and the prosensory domain, whilst Jag1 was almost extinguished in this domain (arrowheads in *Figure 1G–G'*). The segregation of the lateral crista then proceeded in a similar manner to that of the anterior crista, with a progressive loss of Sox2 expression at the interface with the prospective utricle, followed by an increase in the distance separating these two patches (*Figure 1—figure supplement 1*). Double-immunostaining for Sox2 and Prox1, a transcription factor highly expressed in the developing cristae compared to the utricle (*Stone et al., 2003*), provided further evidence that the prospective cristae are specified whilst they are still part of a larger prosensory domain (*Figure 1*, H-J'). Hence, both cristae appeared to form by progressive segregation from a larger prosensory domain (giving rise to the utricle), during which the down-regulation of Jag1 precedes that of Sox2 expression in prospective non-sensory regions. This suggests that Notch activity is dynamically regulated during the early stages of sensory patch specification and their segregation.

## The early Notch-active otic cells contribute to both sensory and non-sensory domains of the differentiated inner ear

To study the dynamics of Notch activity during sensory patch formation, we used in ovo electroporation and two complementary methods to identify and trace the early Notch active cells in the developing chick inner ear. We first used a bicolor fluorescent reporter of Notch activity (*Figure 2A*),

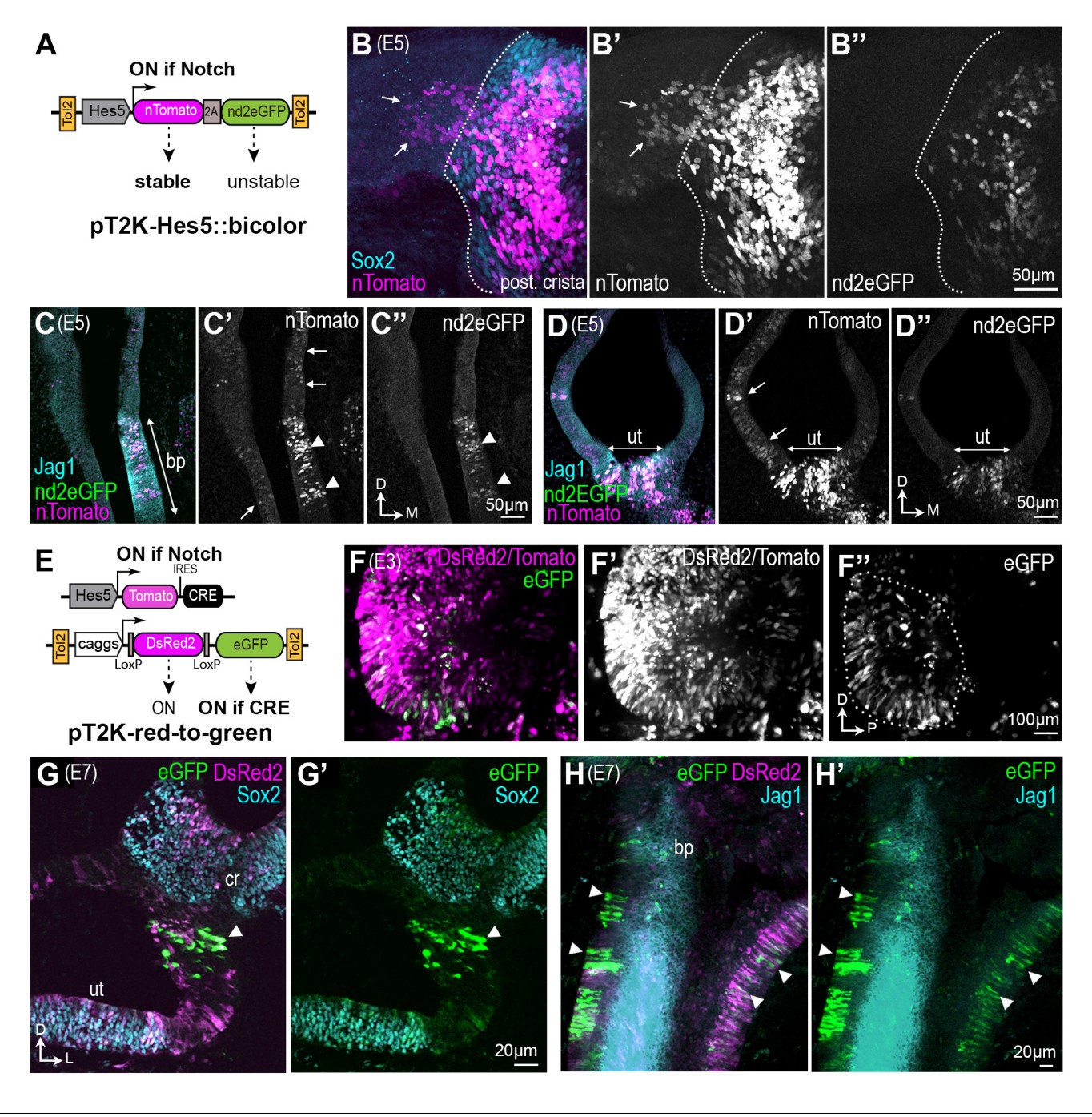

**Figure 2.** Some otic cells turn off Notch activity and adopt a non-sensory fate during sensory patch formation. (A–D'') Identification of 'Notch-OFF' cells using a bi-color Notch-sensitive Hes5 reporter as a molecular timer. (A) The Hes5::bicolor Tol2 construct contains the Hes5 promoter, regulating expression of a stable nTomato and a destabilized nd2eGFP. (B–B'') Posterior crista (whole-mount) analysed 72 hr after electroporation with the Hes5:: bicolor transposon and stained for Sox2 (cyan) to identify sensory progenitor cells. Note the presence of 'Notch-OFF' cells (arrows) with low levels of nTomato (magenta in B; gray in B') but no nd2eGFP fluorescence (green in B, gray in B''), in the non-sensory domain that flanks the ventral border of the crista (dotted line). Transverse section of an E5 cochlear duct (C–C'') and (D–D') utricle transfected with the Hes5::bicolor construct and immunostained for Jag1 (cyan). Notch-active cells (arrowheads) are located within the sensory patches (bp = basilar papilla; ut = utricle). Note the presence of 'Notch-OFF' cells, with nTomato fluorescence only, outside of the sensory patches (arrows in C' and D'). (E–H') Long-term genetic labelling of the early Notch-active cells using Cre-mediated recombination and a red-to-green Tol2 construct. (E) The Hes5::Tomato-IRES-Cre plasmid drives transient expression of the Tomato fluorescent protein and Cre recombinase in Notch-active cells. The pT2K-red-to-green Tol2 transposon drives constitutive expression of either DsRed2, or alternatively eGFP after Cre-mediated recombination. (F) Whole-mount view of an E3 chick otocyst 16 hr

*Figure 2 continued on next page*

*Figure 2 continued*

after co-electroporation with Hes5::Cre and PT2K-red-to-green. The majority of transfected cells are DsRed2 fluorescence only (**F'**), but a subset of cells, located in the anterior domain of the otocyst, exhibit eGFP fluorescence (outline in **F''**). (**G–G'**) Transverse section through the vestibular system of a transfected chick inner ear at E7, immunostained for Sox2 (cyan). In this example, a group of eGFP-expressing cells (arrowhead) are present in the non-sensory domain separating the utricle (ut) from the crista (cr). (**H–H'**) Whole-mount preparation of the cochlear duct of a pT2K-red-to-green transfected chick inner ear at E7, immunostained for Jag1 (cyan). Note the presence of several groups of eGFP-expressing cells in the non-sensory domains surrounding the basilar papilla (arrowheads).

DOI: https://doi.org/10.7554/eLife.33323.005

The following figure supplement is available for figure 2:

**Figure supplement 1.** Representative whole-mount views of chicken otocysts 16 hr after electroporation with different combinations of pT2K-red-to-green and Hes5::Tomato-IRES-Cre constructs.

DOI: https://doi.org/10.7554/eLife.33323.006

Hes5::bicolor, to identify the Notch active (Notch-ON) cells and those in which Notch activity has been switched off, or Notch-OFF cells. This reporter uses the mouse Hes5 promoter, active in Notch-ON cells in the developing chick inner ear (*Chrysostomou et al., 2012*), to drive expression of two fluorescent proteins linked by a 2A peptide sequence (to ensure stoichiometric production): a stable tomato-fluorescent protein fused to Histone 2B (nTomato; persistent for at least 48 hr) and a destabilised and nuclear localized GFP, nd2eGFP, with a half-life of approximately 5 hr after Notch inhibition (data not shown and [*Chrysostomou et al., 2012*]). The reporter cassette was cloned into a Tol2 transposon vector allowing stable integration into the genome of transfected cells after co-electroporation with the Tol2 transposase. We reasoned that cells transfected with this 'molecular timer' would fall into two categories: those in which Notch is active, or has been recently active, would exhibit red and green fluorescence in their nuclei; those in which Notch had been active previously but turned off Notch activity more than 10 hr previously should exhibit nTomato fluorescence only (*Figure 2B–B''*). Following in ovo electroporation at E2, the embryos were incubated until E5-E7, a stage at which distinct sensory domains are apparent. Analysis of successfully electroporated samples revealed that as expected, red/green nuclei belonged to cells located within the Sox2+ sensory domains. However, cells with red (but not green) nuclei were also present outside the sensory domains (*Figure 2B,C*). These cells had lower levels of red fluorescence than those located within the sensory territories, as expected if these cells had been Notch-active at an earlier stage of development, but subsequently switched off Notch activity (*Figure 2B–D*). An alternative interpretation, however, is that these cells were experiencing very low levels of Notch activity at the time of examination, sufficient to elicit an accumulation of the H2B-Tomato protein, but not the destabilised nd2eGFP. Hence, we next used a different strategy to irreversibly label the early Notch-active cells with a 'red-to-green' Tol2 construct (*Hans et al., 2009*). In this construct, a constitutively active promoter drives the expression of either DsRed2, or eGFP following Cre-mediated recombination. Chicken otocysts were electroporated at E2 with the Tol2 pT2K:red-to-green and a Hes5::Tomato-IRES-Cre plasmid (thereafter Hes5::Cre), driving transient Tomato and Cre expression in Notch-active cells (*Figure 2—figure supplement 1*). Examination of otocysts 16 hr post-electroporation showed large numbers of red (DsRed2 and/or Tomato) fluorescent cells, but the eGFP+ cells were observed almost exclusively in the anterior and occasionally posterior regions of the otocyst (5 out of 8 samples; the remaining 3 not showing any eGFP+ cells). The presence of eGFP+ cells in domains where Notch is normally active at this stage confirmed successful recombination of the Tol2-red-to-green transgene in the early Notch-active cells. There were no eGFP+ cells in samples electroporated with the red-to-green construct alone, confirming that no spontaneous recombination occurs in the absence of Cre expression (*Figure 2—figure supplement 1*). In samples allowed to develop until E7 (n = 7) and immunostained for Sox2 and Jag1, the majority of cells exhibited red fluorescence. Strikingly, the eGFP+ cells, derived from the early Notch-active cells, were detected in both sensory and non-sensory territories (*Figure 2G–H'*). Combined with results from our Hes5::bicolor reporter, these data suggest that Notch signalling is down-regulated in a proportion of early Notch-active prosensory cells, which then switch character and differentiate as non-sensory cells.

## A gain of lateral induction leads to defects in the positioning of sensory patch boundaries and to sensory domain fusion

We next asked what would be the consequences of increasing the levels of lateral induction during sensory patch morphogenesis. To investigate this, we stably transfected the developing chick inner ear with a Tol2 transposon DNA construct, pT2K-Hes5::Dll1-eGFP, which induces expression of both Delta-like 1 (Dll1) and eGFP in Notch-active cells (*Chrysostomou et al., 2012*), and therefore the Jag1-expressing prosensory cells (*Figure 3A*). Following *in ovo* electroporation at E2, the inner ear was examined in whole-mount samples collected between E5 and E14. Compared to samples electroporated with the Hes5::d2eGFP construct (*Figure 3B*), we observed varying morphological defects, primarily in the vestibular system, in samples with moderate to high levels of Hes5::Dll1-eGFP fluorescence (n = 21). In 'mild' cases, that is those with moderate Hes5::Dll1-eGFP expression (*Figure 3C*), the size of the vestibular system appeared slightly reduced (compare for example the width of the vestibular system in *Figure 3B and C*) but distinct patches of eGFP fluorescence corresponding to the normal location of the sensory organs were observed (6/21). In the remaining 15 samples, the morphogenesis of the vestibular system was more severely affected, with reduced overall size (*Figure 3D*) and missing semi-circular canals (data not shown). Additionally, we observed abnormal eGFP fluorescence in the dorsal portion of the vestibular system and in canal-like, elongated domains (asterisk in *Figure 3D*), contrasting with the normal appearance of segregated sensory domains in controls (*Figure 3B*). We performed immunostaining for hair cell markers (HCA and otoferlin) and Jag1 to analyse sensory patch morphology in Hes5::Dll1 transfected samples in greater detail. In samples with relatively mild defects in ear morphology analysed at E8-E14, eGFP + cells were few and observed within or in close proximity to easily recognizable sensory organs, such as the cristae of the vestibular system (*Figure 3E–H*). In these samples, groups of ectopic hair cells, intermingled with eGFP+ cells, were occasionally found at the lateral border of sensory organs (*Figure 3E–F*). Alternatively, when eGFP+ cells were in greater numbers, the pattern of hair cell differentiation and sensory organ shape were affected (*Figure 3G–G'*). In samples with more severe defects in overall inner ear morphology, eGFP fluorescence was frequently detected in the dorsalmost region of the vestibular system. Importantly, eGFP expression was associated with an increase in Jag1 expression at E5 (*Figure 3I–I'*) and E8 (*Figure 3J*), indicating that the Hes5::Dll1-eGFP construct was able to promote endogenous, Jag1-mediated lateral induction. At E14, the most severely affected samples showed widespread fusion of the sensory patches and elongated or canal-like eGFP+ domains with ectopic hair cells in the dorsal part of the vestibular system (*Figure 3K–L'*). In regions containing large clusters of eGFP+ cells, hair cell density was drastically reduced (*Figure 3L–L'*), which was an expected consequence of the trans-inhibition of hair cell formation by Dll1, as previously described (*Chrysostomou et al., 2012*).

In summary, the morphological defects induced by a gain of lateral induction are complex and variable in severity. They include abnormal positioning of the boundaries of the sensory patches as well as formation of ectopic or fused sensory territories. This suggests that the precise regulation of lateral induction and perhaps its localized inhibition are essential for normal sensory patch segregation.

## Notch signalling antagonizes cLmx1b expression

Lmx1a, a LIM-homeodomain transcription factor, is essential for the segregation of inner ear sensory patches during development (*Koo et al., 2009*; *Steffes et al., 2012*; *Nichols et al., 2008*). In *Lmx1a* mutant mice, the sensory domains are abnormally fused, a phenotype reminiscent of that observed after the gain of lateral induction in the chick inner ear. This prompted us to investigate the potential interactions between Notch/Jag1 and cLmx1b, the functional homologue of mouse Lmx1a in the chicken inner ear (*Giraldez, 1998*; *Abello and Alsina, 2007*). We found, in agreement with previous studies, that *cLmx1b* is excluded from the anterior neuro-sensory competent patch of the E3 chicken otocyst (*Figure 4A–B''*). Over subsequent stages of development, *cLmx1b* is gradually down-regulated from all prosensory domains as they elevate expression of Jag1, but remains elevated in regions that give rise to the endolymphatic sac and duct, and the non-sensory territories that separate the cristae from the utricle and saccule (data not shown), as described for *Lmx1a* expression in the mouse inner ear (*Koo et al., 2009*; *Steffes et al., 2012*; *Nichols et al., 2008*).

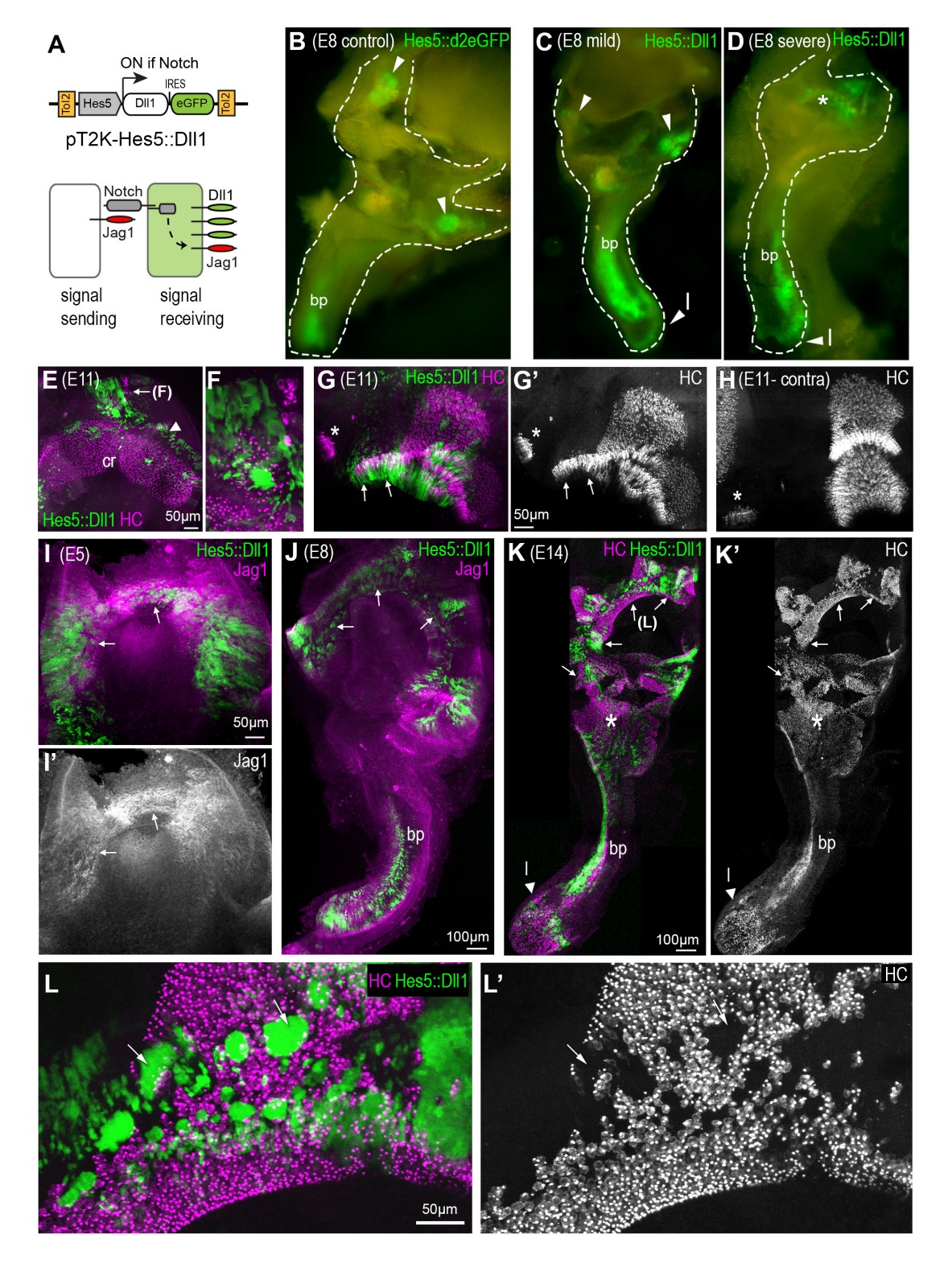

**Figure 3.** A gain of lateral induction disrupts the patterning and boundaries of inner ear sensory patches. (**A**) Schematic of the pT2K-Hes5::Dll1 Tol2 construct used to induce an artificial gain of lateral induction. The Hes5 promoter drives expression of Dll1, along with eGFP, in transfected cells in which Notch is active. (**B–D**) Dissected E8 inner ears transfected with either a control Hes5::d2eGFP (**B**) or a Hes5::Dll1 (**C–D**) Tol2 construct at E2.5. In the control (**B**), eGFP fluorescence is detected in distinct sensory patches, in this case the basilar papilla (bp) and two cristae (arrowheads). Transfection

*Figure 3 continued on next page*

*Figure 3 continued*

with Hes5::Dll1 induces mild (**C**) or severe (**D**) defects in the morphogenesis of the vestibular system, and abnormal activation of eGFP expression in the dorsal regions (asterisk in **D**). (**E–H**) whole-mount preparations of E11 Hes5::Dll1 transfected cristae, immunostained with HCA and myosin 7A antibodies (in magenta) to identify hair cells. Note the formation of ectopic hair cells (**E–F**) or abnormal expansion of the posterior crista (arrows in **G–G'**) towards the macula neglecta (asterisk) compared to the contralateral untransfected posterior crista (**H**). (**I–I'**) Jag1 expression is induced (arrows) in the dorsal portion of an otocyst with strong activation of the Hes5::Dll1 construct. (**J**) Example of an E8 inner ear with ectopic induction of Jag1 and Hes5::Dll1 activity in the dorsal region of the vestibular system.(**K–K'**) Whole-mount of an E14 inner ear with widespread Hes5::Dll1 induction, immunostained with HCA and myosin 7A antibodies (in magenta). The vestibular system contains ectopic and fused sensory domains (arrows). Hair cells are also present in between the basilar papilla (bp) and the saccule-utricle region (asterisk), which appear to form a continuous sensory patch. (**L–L'**) Higher magnification view of the dorsal vestibular region of the sample shown in (**K**). Hair cell density is reduced inside the clusters of transfected cells with strongest eGFP fluorescence (arrows).

DOI: https://doi.org/10.7554/eLife.33323.007

A previous study has shown that pharmacological inhibition of Notch activity in the chick otocyst upregulates and expands *cLmx1b* expression (*Abello and Alsina, 2007*), suggesting that Notch activity could antagonize cLmx1b/Lmx1a expression. To test this hypothesis further, we electroporated E2 chicken otic cups with a plasmid that drives expression of the chick Notch1-intracellular domain (NICD), mimicking a constitutive activation of the Notch receptor (*Daudet and Lewis, 2005*). When compared to the contralateral (left, unelectroporated) ear, *cLmx1b* expression, analysed by in situ hybridisation, was strongly reduced in NICD-electroporated otocysts (n = 6/6)

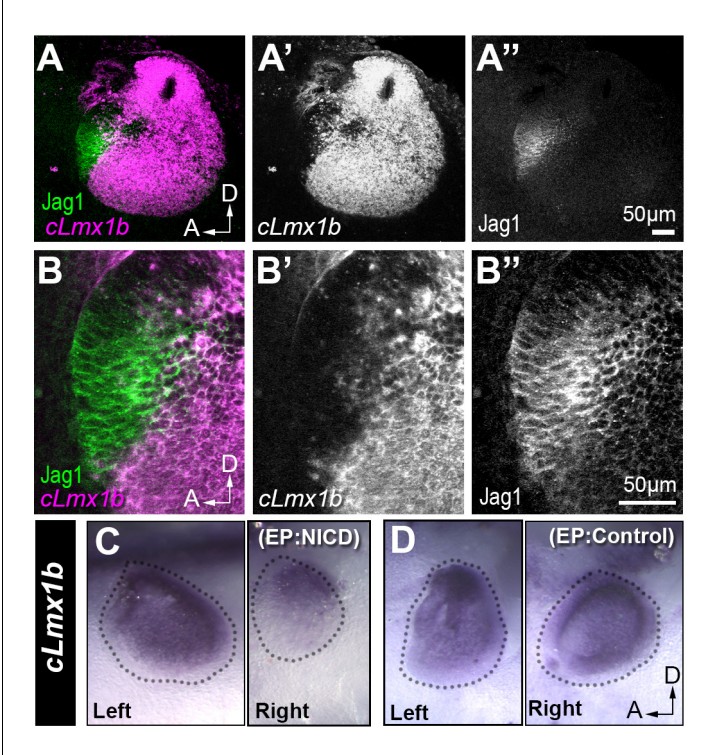

**Figure 4.** Jag1 and cLmx1b are expressed in complementary domains and Notch inhibits cLmx1b expression in the chick otocyst. (**A–B''**) Fluorescent in situ hybridization for *cLmx1b* (magenta) and immunostaining for Jag1 (green) in the E3 chick otocyst. (**A, A''**) Jag1 is expressed in a complementary manner to *cLmx1b*. (**B–B''**) High magnification view of the anterior region of the otocyst shown in (**A**). (**C–D**) Whole-mount in situ hybridization for *cLmx1b* 24 hr after in ovo electroporation with plasmids encoding either NICD (**C**) or a control red fluorescent protein (**D**) used at the same concentration. Note the reduction in *cLmx1b* expression in the NICD-transfected sample (**C**, right) in comparison to the untransfected contralateral otocyst (**C**, left) or to the otocyst transfected with a control plasmid (**D**, right).

DOI: https://doi.org/10.7554/eLife.33323.008

examined 24 hr after electroporation (*Figure 4C*). There was no such down-regulation in any of the otocysts (n > 10) transfected with a control plasmid driving expression of mRFP1 only (*Figure 4D*). These results confirm that Notch activity antagonizes *cLmx1b* expression, and could be at least one of the factors responsible for its progressive exclusion from developing sensory patches.

## Overexpression of cLmx1b inhibits Jag1 and commitment to the sensory fate

We next asked whether cLmx1b conversely impacts Notch signalling and sensory specification. We generated a Tol2 plasmid for long-term and constitutive co-expression of cLmx1b and eGFP (pT2K-Lmx1b-eGFP) in the developing chick inner ear. After in ovo electroporation at E2, inner ear tissue was processed for Jag1 and Sox2 immunostaining between E4 and E7. At both stages, cLmx1b-overexpressing cells, identifiable by eGFP expression, showed reduced levels of Jag1 expression (*Figure 5*). This effect was particularly striking at the lateral borders of the sensory domains at E4.5 (*Figure 5A–A''*) and at E7 (*Figure 5B–D''*). In the case of extensive transfection, very large portions of the Jag1+ sensory epithelia were missing (see for example *Figure 5B–B''* and *Video 1*) and the presence of cLmx1-transfected cells resulted in sensory epithelia with very irregular boundaries. The cLmx1b-overexpressing cells tended to form aggregates, exhibiting frequent apical constrictions and tightly surrounded by Jag1+ cells when located inside a sensory patch (*Figure 5C–D''*). These defects were never observed in samples transfected with a control pT2K-eGFP plasmid (*Figure 5E–E''*). These data suggest that cLmx1b antagonizes the propagation of lateral induction at sensory patch borders by interfering with Jag1 expression.

In contrast to that of Jag1, the expression of the prosensory marker Sox2 was not systematically affected by cLmx1b overexpression (*Figure 6A,C*). At E4.5, the majority of cLmx1b-overexpressing cells located inside the sensory patches retained detectable Sox2 staining in their nuclei (*Figure 6A–B'*). At later stages (E7), the cLmx1b-overexpressing cells fell into two categories: inside the sensory patches, such as the utricle (*Figure 6B,D'*), they tended to retain Sox2 expression; at the border of the sensory patches, we observed clusters of cells devoid of Sox2 expression (*Figure 6C,E–E''*). Hence, the ability of cLmx1b to repress Sox2 expression is context-dependent, and appears most pronounced at the lateral border of developing sensory epithelia.

## Loss of Lmx1a causes an expansion of Jag1 expression in the embryonic inner ear

If cLmx1b expression antagonizes lateral induction during inner ear development, one would predict that Jag1 expression might be elevated under conditions in which its function has been lost. To test this hypothesis, we first examined the inner ear of Belly-Spot and Deafness (*bsd*) mice, which carry a genomic deletion of exon three and produce a truncated and non-functional version of the Lmx1a protein (*Steffes et al., 2012*). We performed immunostaining in the inner ears of E14.5 *bsd* heterozygous (phenotypically normal) and homozygous (null) mice. In heterozygotes, the different sensory organs were easily identifiable and well separated; in the vestibular organs, the highest levels of Jag1 were found inside the sensory cristae (*Figure 7A–B'*). In contrast, the inner ear of *bsd* mutant mice were smaller, with a shorter cochlear duct and an abnormally shaped vestibular system (*Figure 7C–D'*). In the ventral-most portion of the vestibular system, Jag1 was expressed uniformly and at relatively high levels throughout the presumptive saccule, utricle, and posterior crista regions. Furthermore, this Jag1 domain was directly connected to that of the cochlear duct, which was also expanded laterally (arrows in *Figure 7D–D'*). Only a few hair cells (identifiable by Myo7a expression) had differentiated in these samples, indicating that the fused sensory domains found in the *bsd* mutant arise as a consequence of the formation of ectopic, Jag1-expressing sensory progenitors.

## Lmx1a expression is gradually confined to non-sensory territories during sensory patch formation

To extend these findings and to ascertain the origin of the ectopic sensory cells in *Lmx1a* mutant mice, we next utilised a knock-in allele, *Lmx1a^{GFP}*, in which the GFP coding sequence is inserted at the *Lmx1a* start codon, abolishing expression of the LMX1A protein (*Giraldez, 1998*).

In heterozygous (*Lmx1a^{GFP/+}*) mice, development of the inner ear proceeded as normal and GFP expression confirmed previously reported regions of Lmx1a expression. The GFP labelling, however,

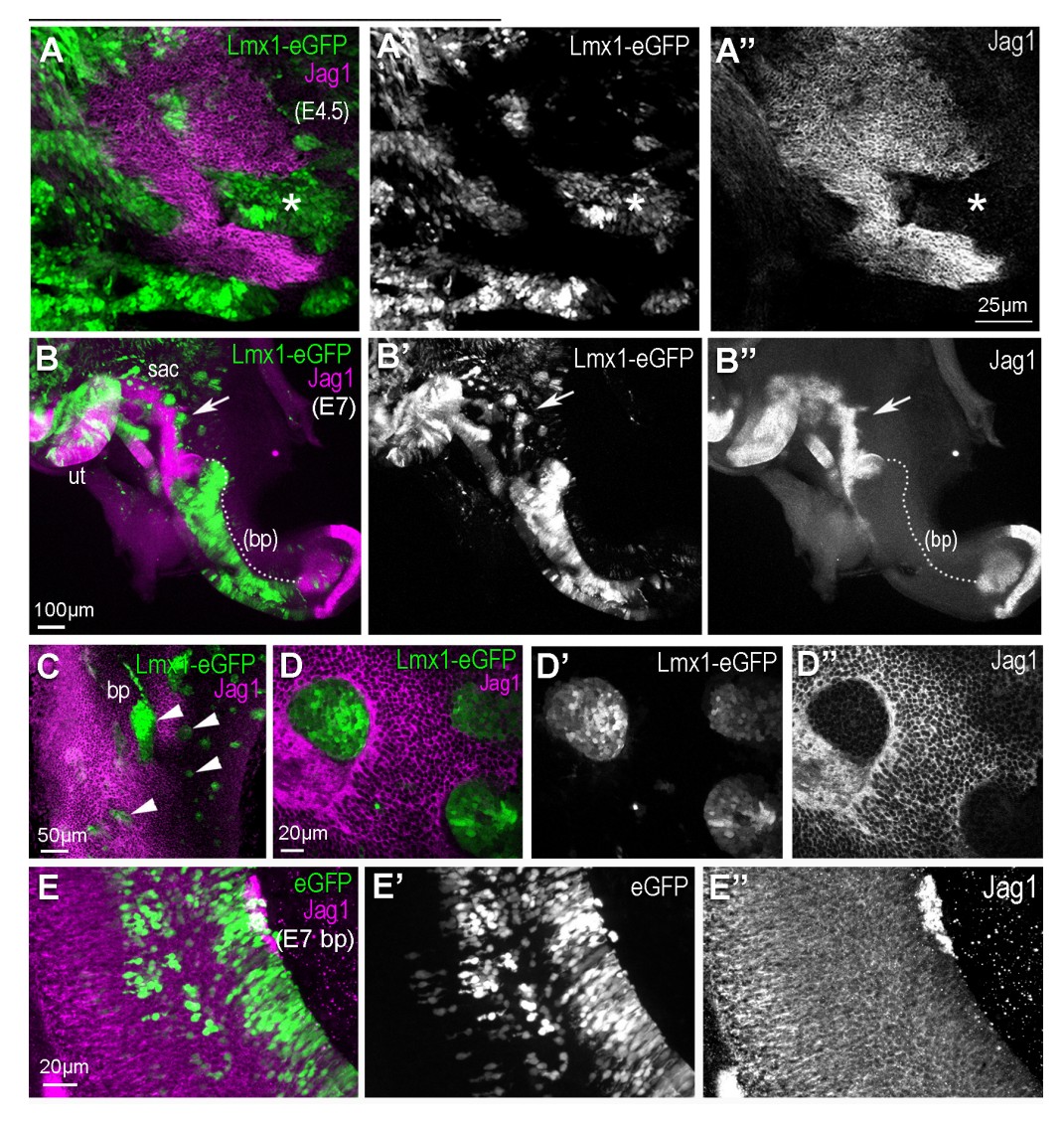

**Figure 5.** Over-expression of cLmx1b down-regulates expression of Jag1. (**A–A''**) Whole mount preparation of an E4.5 chick utricle after electroporation with the pT2K-Lmx1-eGFP Tol2 construct and immunostaining for Jag1. Note the irregular contours of the utricular macula and the reduced expression of Jag1 in transfected cells (asterisk). (**B–B''**) Whole mount view of an Lmx1-eGFP transfected chick inner ear at E7. Jag1 expression is reduced or completely absent in cLmx1b-overexpressing cells, resulting in abnormal positioning of the boundaries of the utricle (ut) and saccule (sac) (arrows in **B'–B''**,) see also **Video 1** for a 3D animation of a high magnification view of this region) and a large truncation (dotted line) of the basilar papilla (bp). (**C**) Surface view of an Lmx1-eGFP transfected E7 basilar papilla with several groups of cLmx1b-overexpressing cells (arrowheads). At higher magnification (**D–D''**), note the rounded appearance of the clusters of transfected cells (**D'**) and the absence of Jag1 expression in those cells (**D''**). (**E–E''**) Surface view of an eGFP (control) transfected E7 basilar papilla. There is no disruption in the expression pattern of Jag1 or the positioning of its lateral borders.

DOI: https://doi.org/10.7554/eLife.33323.009

permitted precise determination of the relationship between Lmx1a, Sox2 and Jag1 expression during mouse inner ear development. At E8 (**Figure 8A** and **Figure 8—figure supplement 1**), GFP was present throughout the otocyst but it was almost completely absent from the anterior and posterior prosensory domains, expressing Sox2 and Jag1. The cells at the edge of the prosensory domains and in the ventro-medial domain of the otocyst were however frequently GFP+. At E10.5–11, the prospective utricle expressed Sox2 and Jag1, but exhibited no or low levels of GFP fluorescence (**Figure 8C–C''** and **Figure 8—figure supplement 1**). The posterior border of this domain was

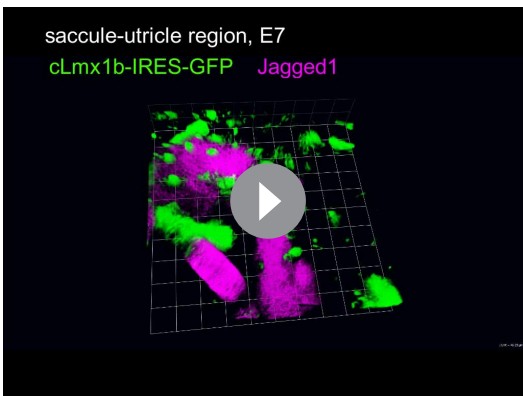

**Video 1.** 3D Velocity rendering of an E7 chicken inner ear electroporated at E2 with cLmx1b-eGFP and immunostained for Jag1 expression (magenta). DOI: https://doi.org/10.7554/eLife.33323.010

directly abutting GFP+ cells which were relatively well aligned. More ventrally, an additional domain of Sox2 and Jag1 expression, presumably corresponding to the prospective saccule, exhibited low levels of GFP fluorescence. In between the prospective saccule and utricle, some cells with reduced Sox2 expression but elevated GFP levels were present. In the E12.5 utricle (*Figure 8D–D''*), Sox2+ and GFP- cells directly abutted the GFP+ cells at the ventro-posterior edge; however, cells with reduced GFP and low or no Sox2 expression were present at the utricle border facing the developing anterior and lateral cristae. At the edges of both cristae and within the saccule, some of the cells were Sox2+/GFP+. In summary, these data show region-specific differences in the relationship between Sox2/Jag1 and Lmx1a expression. At the posterior edge of the prospective utricle, Sox2/Jag1-expressing cells and GFP+ cells were separated by a relatively smooth boundary, suggesting limited cell mixing or changes in gene expression throughout development. In contrast, the domains of the utricle facing the cristae and the saccule exhibited

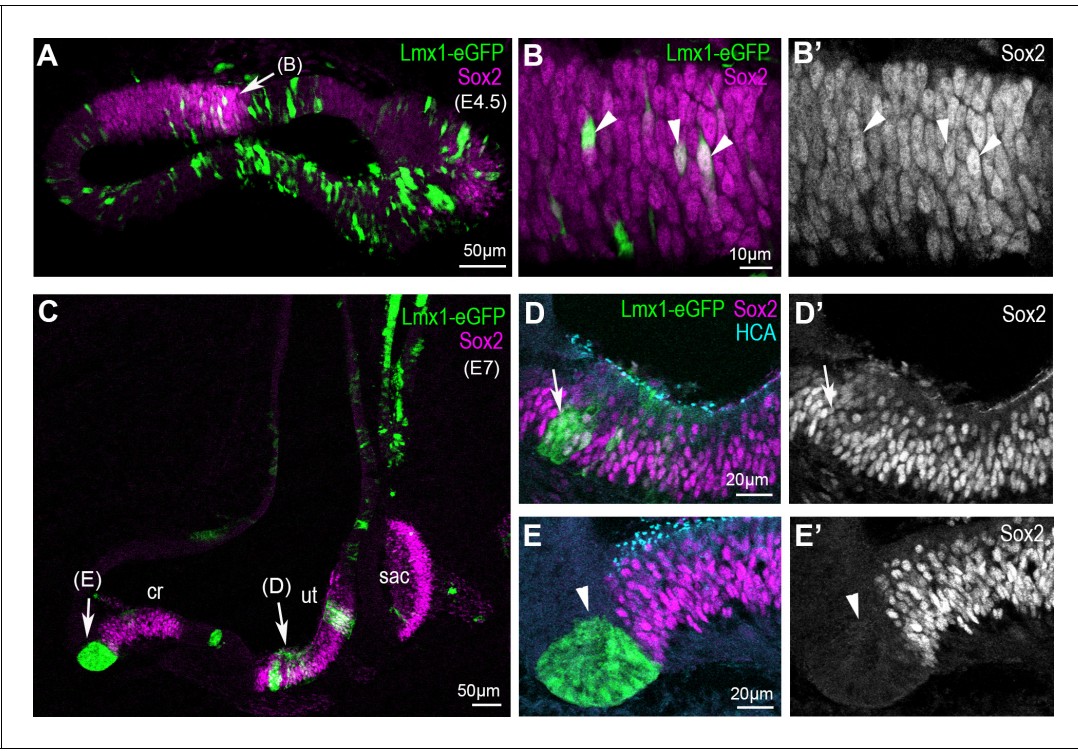

**Figure 6.** Overexpression of cLmx1b down-regulates Sox2 expression in a context-dependent manner. (A–E') Transverse views of the developing chick inner ear following electroporation with pT2K-Lmx1-eGFP and immunostaining for Sox2 (magenta) and HCA (cyan). (A) An E4.5 sample with two vestibular sensory patches containing transfected cells. At high magnification (B–B''), note that the levels of Sox2 are unchanged in cLmx1b-overexpressing cells (arrowheads) compared to neighbouring untransfected cells (B–B'). (C) Low magnification view of an E7 sample, with the utricle (ut), saccule (sac) and anterior crista (cr) visible. (D–D') High magnification view of the utricle, containing cLmx1b-overexpressing cells that retain Sox2 expression (arrows), although at a reduced level compared to neighbouring untransfected cells. (E–E') A cluster of transfected cells is abutting the anterior crista, and show a complete absence of Sox2 expression. Note also the apical constriction of the transfected cells (arrowheads).
DOI: https://doi.org/10.7554/eLife.33323.011

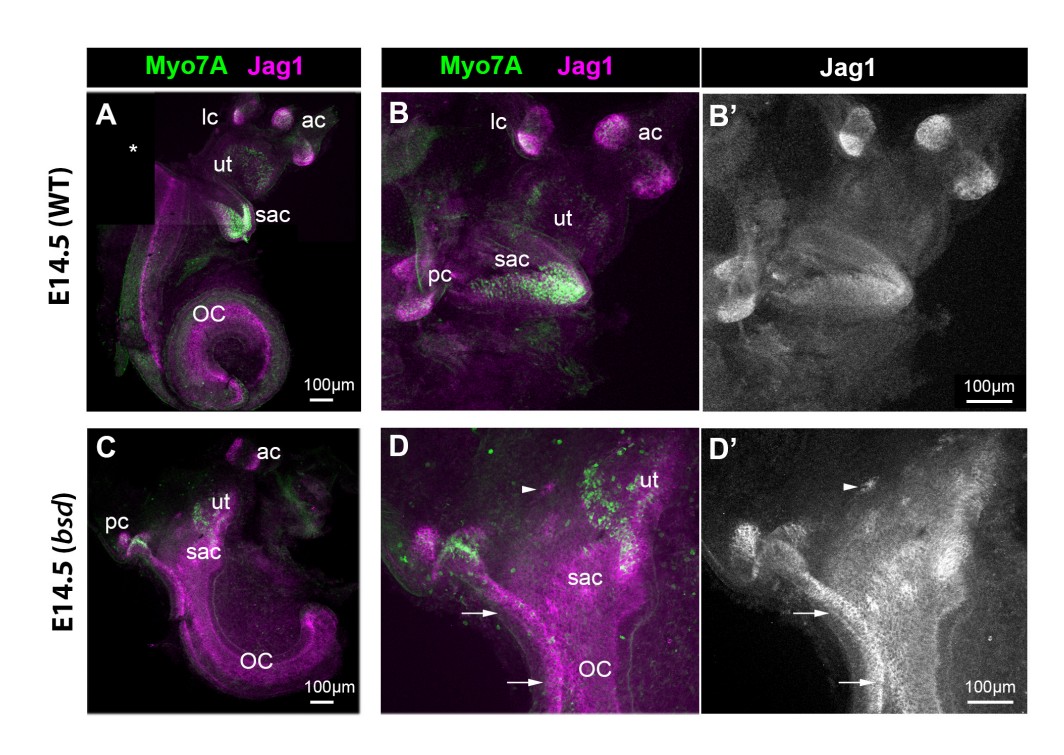

**Figure 7.** Jag1 expression is expanded in the inner ear of E14.5 *bsd* mutant mice. Whole-mount preparations of the inner ear of E14.5 wild-type and *bsd* mice, immunostained for Jag1 and the hair cell marker myosin 7A (Myo7A). In the wild-type (**A–B'**), several patches of jag1 expression corresponding to the distinct vestibular organs and the organ of Corti are visible. Highest levels of expression are found in the cristae. In the *bsd* mutant (**C–D'**), Jag1 expression is expanded throughout the vestibular system, with occasionally patches of cells exhibiting higher expression levels (arrowhead in **D–D'**). Individual patches are difficult to identify, with the exception of the anterior and posterior cristae due to their position and partial segregation. Note the continuity in Jag1 expression between the posterior crista and the organ of Corti (arrows in **D–D'**) and the absence of segregation between the saccule and utricle domains. Abbreviations: lateral (lc), anterior (ac) and posterior (pc) cristae; utricle (ut); saccule (sac); organ of Corti (OC).

DOI: https://doi.org/10.7554/eLife.33323.012

some overlap or mixing between Sox2/Jag1+ and Lmx1a/GFP+ cells. This suggests that dynamic changes in the expression of these genes are associated to the segregation of the cristae and saccule from the utricle.

## Lmx1a is required for the normal differentiation of non-sensory cells

We next compared the inner ear morphology of *Lmx1a*^{GFP/+} and *Lmx1a*^{GFP/GFP} littermates at postnatal day 1 (P1) and found that this knock-in allele recapitulates the previously reported *Lmx1a* mutant phenotypes (*Koo et al., 2009*; *Steffes et al., 2012*; *Nichols et al., 2008*). In heterozygous P1 mice (*Figure 9A–D*), inner ear morphology was normal and strong GFP expression was present at known sites of *Lmx1a* expression, such as the lateral wall of the organ of Corti (*Figure 9A,C–D*), the endolymphatic duct (not shown), and the various non-sensory tissues surrounding the vestibular sensory epithelia (*Figure 9B*). The lateral compartment of the organ of Corti, composed of the Claudius and the Hensen cells, exhibited lower levels of GFP fluorescence compared to the lateral wall of the cochlea (*Figure 9C–D*). The *Lmx1a*^{GFP/GFP} P1 mice had a cyst-like vestibular region without semi-circular canal or endolymphatic duct, and a shortened cochlear duct (*Figure 9E*). In the vestibular system, the sensory patches were misshapen and fused in some locations, but GFP-expressing territories were still present in between adjacent sensory patches (*Figure 9F*). Additional hair cells rows were present in the basal most turn of the cochlea, the numbers of which varied between littermates (*Figure 9G*). The expression of Jag1 remained confined to the sensory territories containing hair cells, and was not expanded medially or laterally. However the cells adjacent to the Sox2

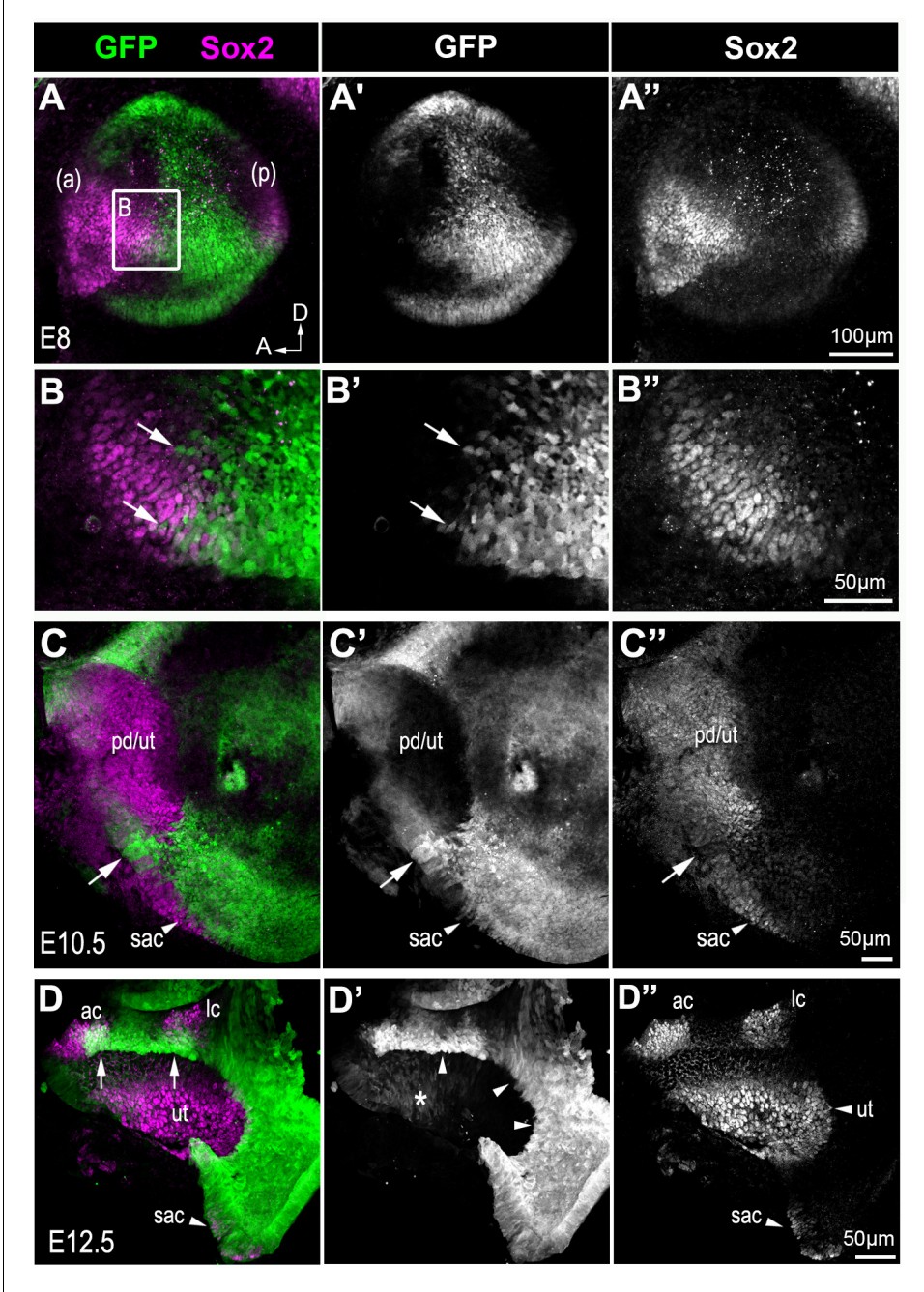

**Figure 8.** Sox2 expression in the developing inner ear of *Lmx1a^{GFP/+}* mice. Whole-mount preparations of the developing inner ear of *Lmx1a^{GFP/+}* mice. At E8, Sox2 expression is present in the anterior (a) and posterior (p) prosensory domains of the otocyst, which also exhibit reduced Lmx1a/GFP expression (**A–A''**). At higher magnification, cell mixing between GFP+ and Sox2+ cells (arrows in **B–B''**) is visible at the posterior border of the anterior prosensory domain. At E10.5 (**C–C''**), Sox2 expression is present in a large anterior prosensory domain (the prospective utricle, labelled pd/ut) and more ventrally in a group of cells that could correspond to the prospective saccule (sac). Some Lmx1a/GFP cells with reduced Sox2 expression are present in between the utricle and the saccule (arrows). By E12.5 (**C–C''**), Sox2 is expressed in distinct sensory patches in the vestibular system, while GFP is present in between the sensory patches. In the utricle (ut), residual GFP expression is present in the anterior-most domain (star in **C'**) and in the cells facing the cristae; at the posterior edge of the utricle, Sox2+ cells are GFP-negative, and directly abut the non-sensory domain with strongly GFP+ cells (arrowheads in **C'**). Note the

*Figure 8 continued on next page*

*Figure 8 continued*
overlap between Sox2 and GFP expression at the borders of the anterior (ac) and lateral (lc) cristae (arrows in **C**) and in the saccule region (sac).
DOI: https://doi.org/10.7554/eLife.33323.013
The following figure supplement is available for figure 8:

**Figure supplement 1.** Jag1 expression in the developing inner ear of *Lmx1a^{GFP/+}* mice.
DOI: https://doi.org/10.7554/eLife.33323.014

+ domain at the lateral border of the organ of Corti had higher levels of GFP fluorescence compared to heterozygous *Lmx1a^{GFP/+}* littermates (compare *Figure 9C–D and G–H*), suggesting an abnormal differentiation of the Claudius and Hensen cells. In the vestibular system of *Lmx1a^{GFP/+}* P1 mice, the non-sensory territory separating the utricle from the anterior and lateral crista was GFP+ and devoid of hair cells (*Figure 9I–I'*). In comparison, this non-sensory domain was reduced in width in *Lmx1a^{GFP/GFP}* samples, and ectopic GFP- hair cells were found intermingled with GFP+ cells at this location (*Figure 9J–J'*). This suggests that in the absence of Lmx1a function, some of the cells that would normally form the (non-sensory) boundary between the utricle and the crista might be diverted towards a sensory fate.

To explore this further, we analysed the fate of the cells from the Lmx1a lineage at earlier stages of development in *Lmx1a^{GFP/+}* and *Lmx1a^{GFP/GFP}* mice (*Figure 10*). In the vestibular system of E15 *Lmx1a^{GFP/+}* mice (*Figure 10A–A''*), GFP+ cells were confined to the non-sensory domains separating Sox2+ sensory patches. In contrast, the vestibular system of *Lmx1a^{GFP/GFP}* mice contained expanded and abnormally shaped Sox2+ and GFP+ domains containing hair cells (*Figure 10B–C''*). A similar overlap between GFP and Jag1 expression was observed in the vestibular system of E14 samples (*Figure 10—figure supplement 1*), confirming that the ectopic sensory cells in *Lmx1a* null animals are derived from *Lmx1a*-deficient cells. Some of the abrupt boundaries between sensory (GFP-) and non-sensory (GFP+) territories were however maintained (arrowhead in *Figure 10B''*). In the cochlear duct of E15 *Lmx1a^{GFP/+}* mice (*Figure 10D–D''*), there was a very abrupt boundary between GFP + and GFP- cells on the medial side of the organ of Corti. On the lateral side there was a more progressive reduction in GFP fluorescence and some overlap in Sox2/GFP+ cells. Since this overlap was not seen in neonatal mice, this suggests a developmental down-regulation of Lmx1a expression in the lateral domain of the organ of Corti. In *Lmx1a^{GFP/GFP}* cochlea, the size of the Sox2+ domain was occasionally expanded (*Figure 10E–E''*), and there were also striking differences in the pattern of GFP fluorescence: at the medial border of the organ of Corti, the abrupt boundary between GFP + and GFP- cells was disrupted and GFP+ cells were intermingled with Sox2+ cells; at the lateral border, the gradient of GFP fluorescence (from high in lateral wall, to low in the Sox2+ domain) was irregular, with clusters of GFP+ cells surrounded by GFP- cells in the Sox2+ domain (arrow in *Figure 10E*). Altogether, these data suggest that the absence of Lmx1a function can lead, in a context-dependent manner, to the abnormal differentiation of cells from the Lmx1a lineage into sensory progenitors. The abnormal pattern of GFP fluorescence at the borders of the organ of Corti could reflect this abnormal differentiation as well as an abnormal cell mixing at sensory patch boundaries.

## Cells at the edge of the utricle display prolonged Lmx1a expression.

We showed previously that GFP fluorescence diminishes progressively at the lateral border of the organ of Corti in *Lmx1a^{GFP/+}* mice, suggesting that cells from the Lmx1a-lineage can adopt a sensory fate in this domain. Another, and perhaps more striking, example of this process was seen at the border of the utricular macula facing the anterior and lateral cristae (*Figure 11*). In samples collected at E18, the border of the Sox2+ domain contained cells with moderate to low levels of GFP fluorescence, intermingled with GFP- cells. Some of these GFP+ cells were also Sox2+, indicating their prosensory character. At P5, a similar but more restricted overlap between GFP+ and Sox2+ cells was noted. At P42, a stage when the utricle is considered fully mature, only a few GFP+ cells were present in the Sox2+ domain, and a mosaic pattern of GFP fluorescence was observed in the transitional cell region. The prolonged mosaicism in GFP fluorescence suggests the existence of a non-lineage restricted tissue boundary, where uncommitted cells can switch fate (in this case lose Lmx1a expression and acquire a sensory character) in response to cues from their external environment.

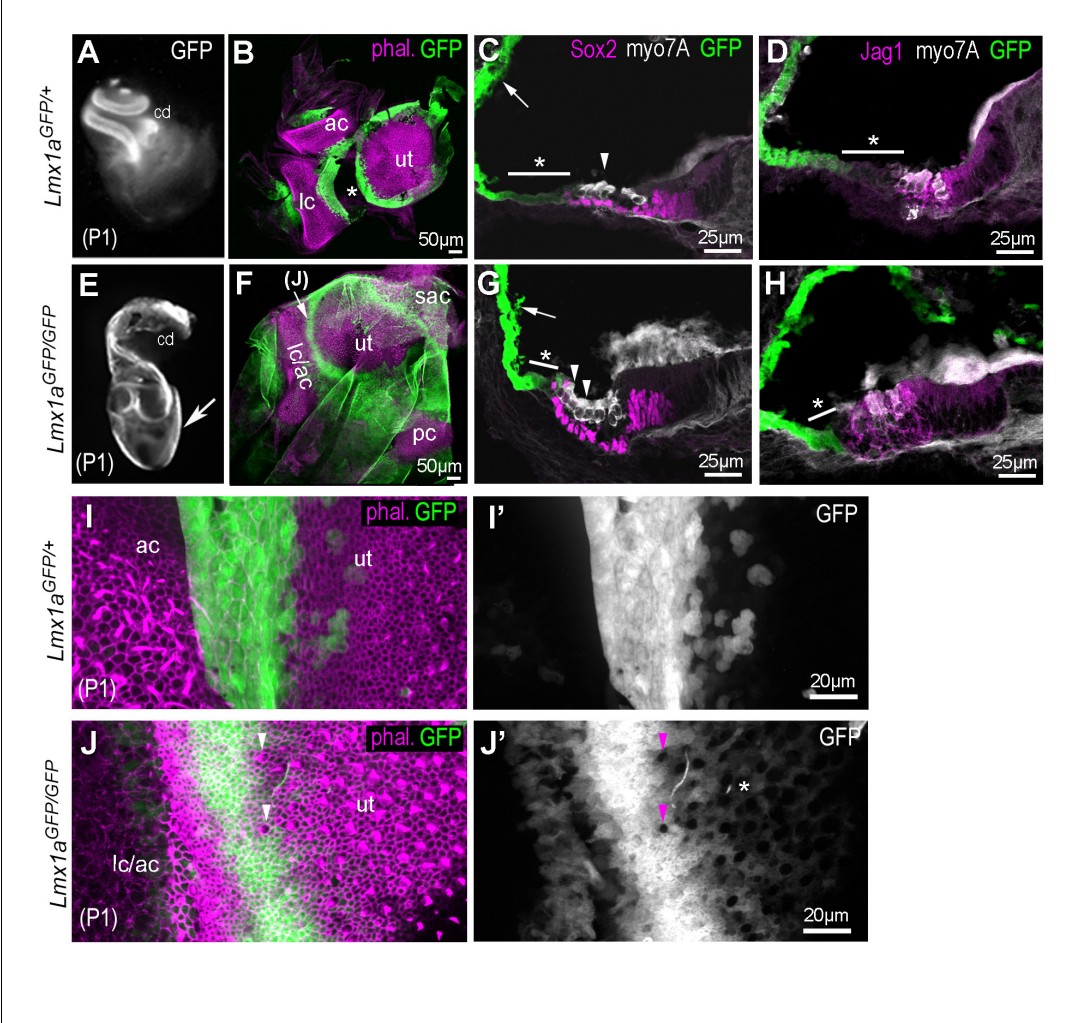

**Figure 9.** Inner ear morphogenesis is severely disrupted in homozygous *Lmx1a^{GFP/GFP}* knock-in mice. In heterozygous P1 *Lmx1a^{GFP/+}* (A–D), inner ear morphology is normal and GFP expression is found at known sites of Lmx1a expression: in the lateral wall of the cochlea (A, arrow in C), in the domains separating the utricle (ut) from the anterior (ac) and lateral (lc) cristae (B; asterisk indicate artefactual interruption of GFP due to dissection). (C–D) Transverse section through the cochlear duct of P1 *Lmx1a^{GFP/+}*mice. The lateral domain of the organ of Corti (asterisk in C–D), composed of the Hensen and Claudius cells, exhibit reduced levels of GFP fluorescence compared to the lateral wall. In the *Lmx1a^{GFP/GFP}* double knock-in mice (E–H), inner ear morphology is severely disrupted: the vestibular system forms a cyst-like structure exhibiting high levels of GFP fluorescence (arrow in E) and the cochlear duct is reduced in size. (F) Some GFP-expressing domains are still present around the utricular macula (ut), the fused anterior and lateral cristae (lc/ac) and the posterior crista (pc) regions. (G–H) The organ of Corti contains additional rows of Myo7a-positive hair cells in the lateral domain (arrowheads in G), but the expression of Sox2 and Jag1 remains confined to the GFP-negative domain. The region abutting the lateral border of the sensory domain (asterisk in G–H) exhibits lower levels of GFP than the lateral wall (arrow in G), but is smaller in comparison to that of the *Lmx1a^{GFP/+}* cochlea (C–D). (I–J') higher magnification surface view of the region separating the utricle from the lateral crista in *Lmx1a^{GFP/+}* (I–I') and in *Lmx1a^{GFP/GFP}* (J–J') mice. Hair cells are identifiable by their actin-rich bundle of stereocilia. In the *Lmx1a^{GFP/GFP}* vestibular system, some hair cells devoid of GFP fluorescence (arrowheads in J–J') are present inside the GFP-positive domain in between the utricle and the lateral/anterior cristae region. Note also the presence of weakly GFP fluorescent cells intermingled with hair cells in the lateral part of the utricle (asterisk in J').
DOI: https://doi.org/10.7554/eLife.33323.015

This mosaicism was more pronounced at the border facing the crista than at other locations of the utricle, suggesting that distinct types of sensory/non-sensory tissue boundaries are present within this organ.

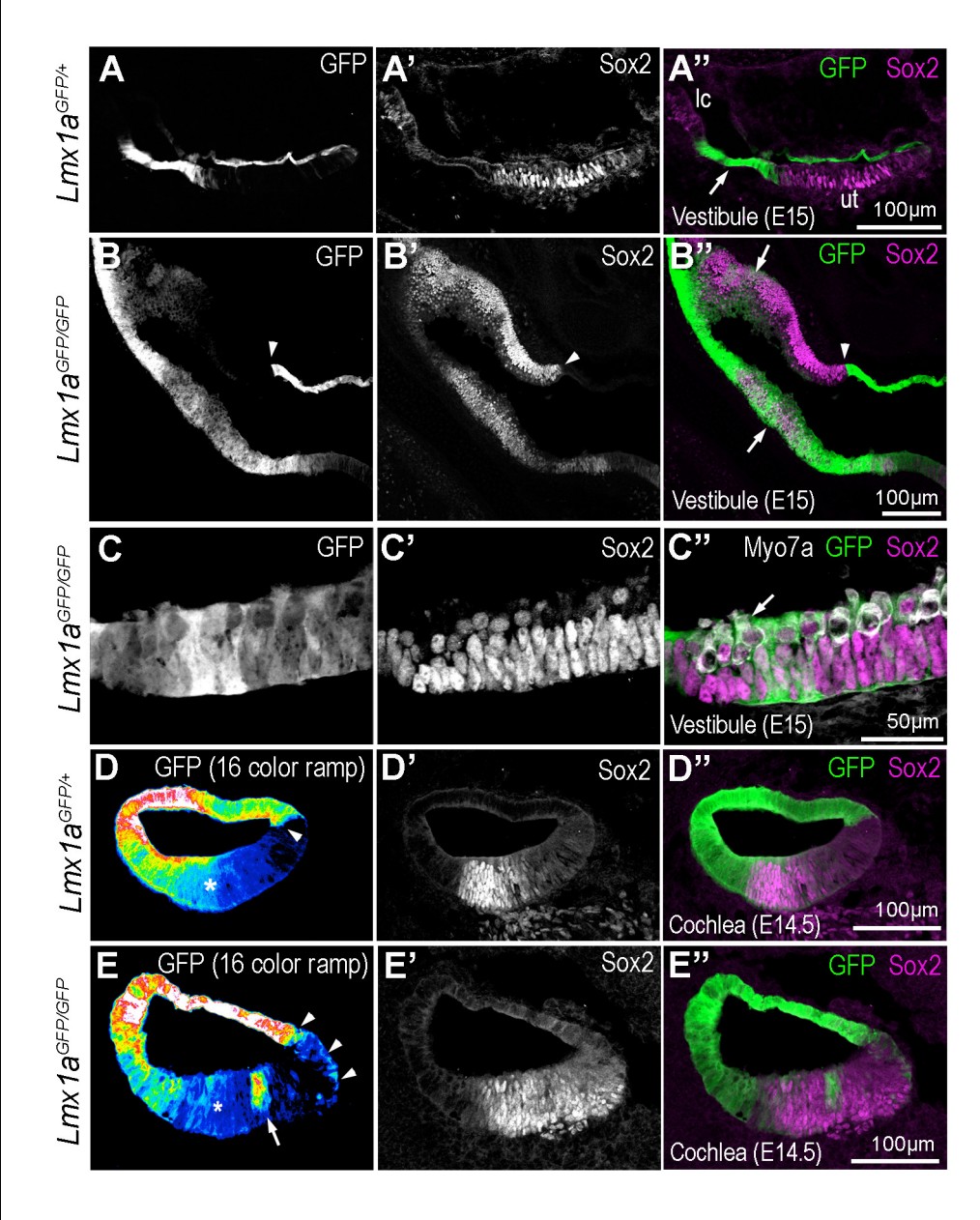

**Figure 10.** Some cells of the Lmx1a lineage are diverted towards a sensory fate in the absence of Lmx1a function. Transverse section through the vestibular system and cochlear duct of embryonic *Lmx1a^GFP/+* and *Lmx1a^GFP/GFP* mice, immunostained for Sox2 and myosin 7A. (A–A'') cross-section through the vestibular system of E15 *Lmx1a^GFP/+* samples. The GFP-positive cells are located in between the utricle (ut) and lateral crista (lc) expressing Sox2, and in the roof of the utricle. In *Lmx1a^GFP/GFP* littermates (**B–C''**), Sox2 expression is expanded but the identity of vestibular patches is difficult to ascertain. At least some of the boundaries between Sox2+ and Sox2- domains coincide with that of GFP expression (arrowheads in **B–B''**). On the other hand, there is some overlap between GFP and Sox2 expression (arrows in **B''**) in some sensory territories. (**C–C''**) High magnification view of a vestibular patch with expression of GFP and Sox2. Some hair cells (arrows) are also present. (**D–D''**) Sox2 expression in the cochlear duct of an E14.5 *Lmx1a^GFP/+* mouse. At the medial border of the developing organ of Corti (arrowhead), cells with either high or low levels of GFP fluorescence (displayed as a 16-color ramp in **D–E**) form a clear interface. At the lateral border, GFP fluorescence decreases progressively, with high levels in the lateral wall and low levels within the Sox2-positive prosensory cells (asterisk). In *Lmx1a^GFP/GFP* littermates (**E–E''**), the lateral gradient of GFP fluorescence is irregular (asterisk) and some cells with fairly high levels of GFP fluorescence are present inside the Sox2-positive domain (arrow in **E**). At the medial border, GFP cells are also intermingled with GFP-negative cells (arrowheads in **E**).

*Figure 10 continued on next page*

*Figure 10 continued*

DOI: https://doi.org/10.7554/eLife.33323.016

The following figure supplement is available for figure 10:

**Figure supplement 1.** Jag1 expression in the vestibular system of E14 *Lmx1a*<sup>GFP/+</sup> and *Lmx1a*<sup>GFP/GFP</sup> mice.

DOI: https://doi.org/10.7554/eLife.33323.017

## Discussion

The adult inner ear exhibits an elaborate three-dimensional structure, with distinct sensory organs interspaced by non-sensory regions. In this study, we explored the mechanisms that control the patterning of these sensory and non-sensory territories and position their boundaries in the chicken and mouse inner ear. Our results support the idea that at least some of the sensory organs and the non-sensory cells that separate them derive from a common pool of sensory-competent cells by segregation. This process is regulated by the balance of opposing signals that either promote (Notch) or prevent (Lmx1) their commitment to the sensory fate (*Figure 12*). We discuss the significance of our findings in relation to existing models for sensory organ specification and some of the implications of the labile character of the early otic sensory progenitors.

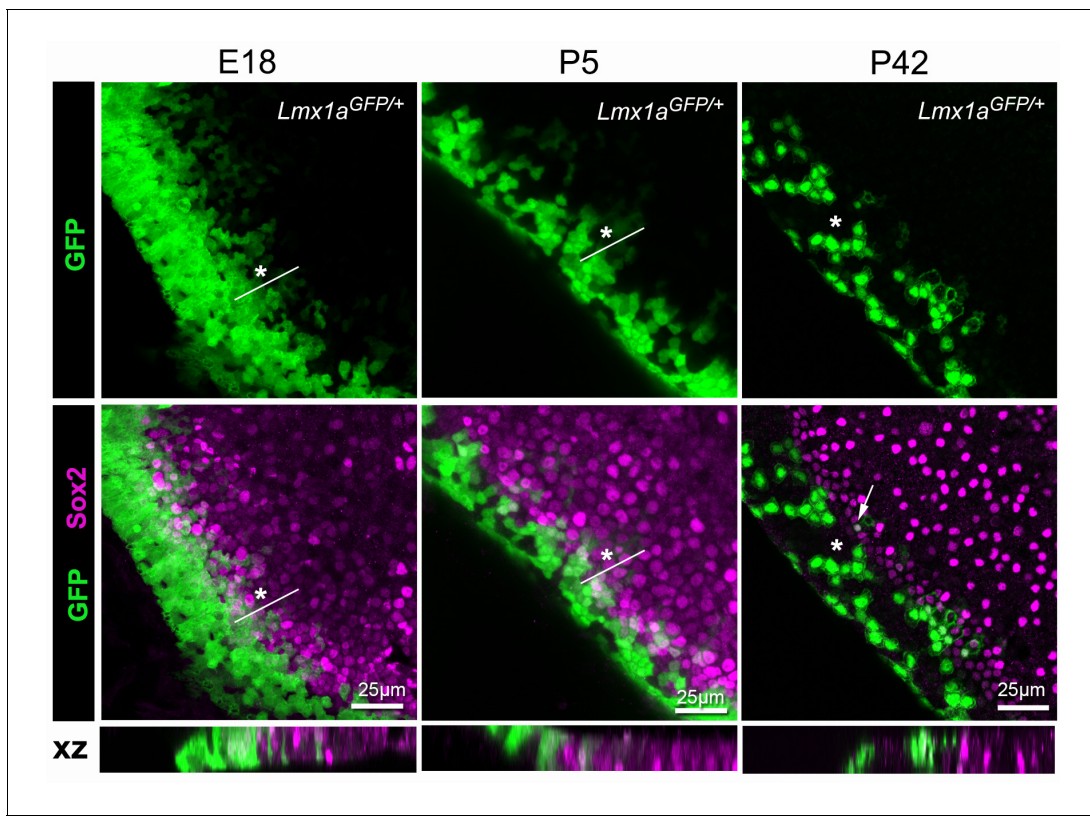

**Figure 11.** Lmx1a expression is progressively down-regulated at the lateral border of the developing utricle. Maximum intensity z-projections of whole-mount utricles from *Lmx1a*<sup>GFP/+</sup> mice at different developmental ages (**E18, P5 and P42**), immunostained for Sox2 expression (magenta). Note the mosaic pattern of GFP fluorescence at the lateral border of the utricle, and the overlap between GFP+ and Sox2+ cells at E18 and P5 (asterisk). In the adult (**P42**) utricle, only a few cells retained low levels of GFP fluorescence and Sox2 expression (arrow) in the sensory domain, whilst mosaic expression of GFP is still observed in the transitional cell region (asterisk). The XY images are orthogonal projections taken from the same preparations.

DOI: https://doi.org/10.7554/eLife.33323.018

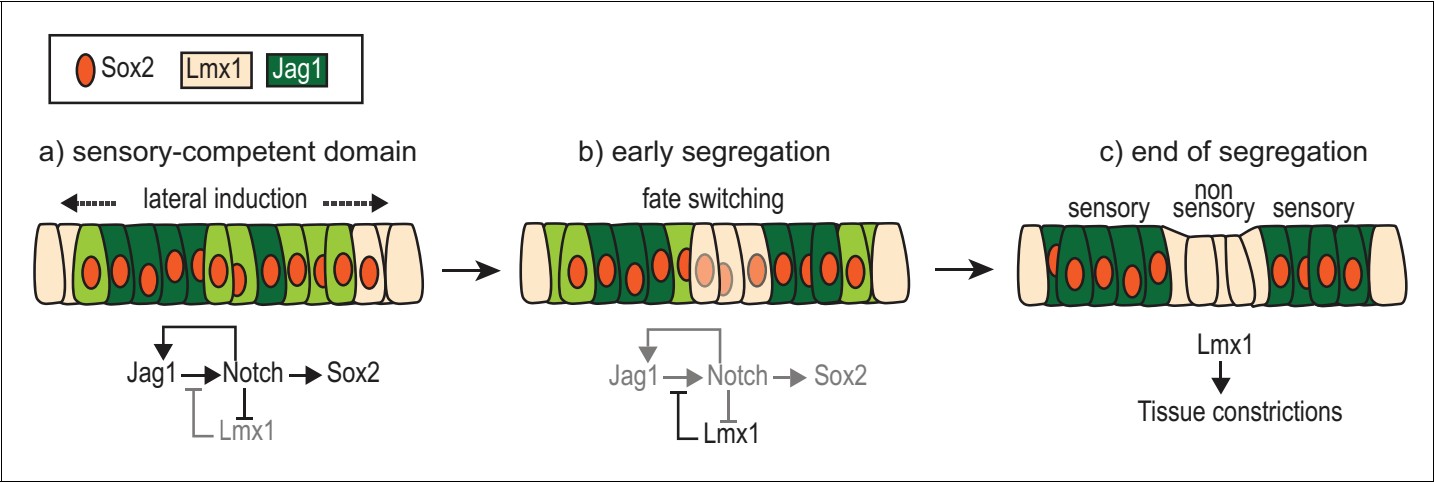

**Figure 12.** A model of the interactions between Notch signalling and Lmx1a during sensory patch segregation. Initially (**a**), the cells of the 'pan' sensory-competent domain express Sox2 as well as Jag1 (strong = dark green or low = light green), which promotes its own expression and adoption of the prosensory fate (and therefore Sox2 expression) by lateral induction. Within the pan-sensory domain, Lmx1a expression is repressed by Notch activity. At its border, a dynamic competition takes place between lateral induction/prosensory specification and the factors promoting Lmx1a expression. (**b**) The segregation of a prosensory domain could be due to a localized reduction in lateral induction within the pan-sensory domain; this could gradually lead to an upregulation of Lmx1 and a conversion (fate switching) of sensory-competent cells into non-sensory cells at the interface of segregating prosensory patches. (**c**) As cells commit to their definitive fate, Lmx1 could lead to the formation of the tissue constrictions that separate adjacent sensory organs. The regulatory interactions depicted are likely to involve intermediate factors.
DOI: https://doi.org/10.7554/eLife.33323.019

## Antagonistic interactions between lateral induction and Lmx1a regulate sensory patch segregation

The morphology of the inner ear and the number of sensory patches it contains vary considerably across vertebrates. In fish and amphibians, it has long been proposed that sensory organs arise by segregation from one or several sensory-competent domains (*Fritzsch et al., 2002*). Histological studies led Knowlton (*Knowlton, 1967*) to suggest that a similar process might occur in the chicken inner ear, in which most sensory epithelia seem to arise from the 'thickened' ventromedial wall of the otocyst. Several studies have shown that the prosensory markers Sox2 and Jag1 are initially expressed in broad territories of the early mouse and chicken otocyst before they become restricted to distinct sensory patches (*Neves et al., 2007*; *Cole et al., 2000*; *Neves et al., 2011*; *Nichols et al., 2008*; *Adam et al., 1998*; *Sánchez-Guardado et al., 2013*), but whether these changes related to sensory patch segregation was unclear. Here, using whole-mount surface preparations of the chicken otocyst, we provide strong evidence that both the anterior and lateral cristae arise from the edge of a large antero-medial Sox2-positive 'pan-sensory' domain, which itself gives rise to the utricle. During this process, Jag1, then Sox2 expression is progressively downregulated at the interface of the cristae and the prospective utricle as the distance between these patches increase.

We propose that lateral induction mediated by Jag1/Notch signalling plays an important role in regulating sensory patch segregation. The defining feature of lateral induction is the intercellular positive feedback loop linking Notch activation to Jag1 expression (*Lewis, 1998*), which could in theory regulate the shape or size of the sensory patches by propagating Notch activity across interacting cells. We tested this hypothesis by increasing the levels of lateral induction in the early embryonic chick inner ear. In contrast to previous studies, which used overexpression of an active form of Notch (NICD) or Jag1 using constitutive or inducible promoters (*Pan et al., 2010*; *Daudet and Lewis, 2005*; *Hartman et al., 2010*; *Neves et al., 2011*), we used a conditional, Notch-responsive promoter (Hes5) to drive the expression of Dll1. By doing so, an additional Notch ligand, Dll1, was induced in the early Notch-active cells, which are the neural or sensory competent progenitors. The developmental defects resulting from Hes5::Dll1 transfection were complex and included: abnormal morphogenesis of the vestibular system, irregular sensory patch boundaries, hair cell patterning

defects, smaller sensory epithelia, and induction of ectopic hair cells. Such variety of defects likely results from a combination of factors: the mosaicism and variability of transfection, the dynamic (Notch-regulated) mode of Dll1 overexpression, the differential response of transfected cells to Notch activation, and the inhibitory function of Dll1 on hair cell differentiation (*Chrysostomou et al., 2012*). Nevertheless, the generation of patches with irregular boundaries and of fused sensory domains strongly suggests that lateral induction regulates the positioning and size of sensory organs. Moreover, the fact that Dll1 can promote ectopic sensory cell formation suggests that the early prosensory function of Notch is not specifically linked to Jagged ligands, but to the induction (or maintenance) of Notch activity. This is consistent with the fact that i) NICD overexpression is able to induce ectopic sensory patch formation (*Pan et al., 2010*; *Daudet and Lewis, 2005*; *Hartman et al., 2010*), and ii) the absence of Dll1, expressed in the neurosensory competent domain during otic neurogenesis, induces defects in sensory patch morphogenesis (*Brooker et al., 2006*). Nevertheless, Dll1 and Jag1 could activate Notch receptors differently, and some evidence suggests that Dll1 induces stronger Notch signalling than Jag1 in the inner ear (*Petrovic et al., 2014*), possibly due to modifications of the Notch receptor by Fringe proteins (*Basch et al., 2016*). Whilst this is advantageous for our 'gain of lateral induction' experiments, it will be important to elucidate the precise molecular mechanisms that fine-tune the signalling abilities of Notch ligands in the ear.

Another implication of these gain-of-function experiments is that lateral induction must be dampened in some of the prospective non-sensory cells to permit sensory patch segregation. This downregulation of Notch activity seems to occur during normal development. In fact, our tracing experiments suggest that some of the early-Notch active cells can switch off Notch activity and develop into non-sensory cells into a wide range of locations. Furthermore, a study in the zebrafish otic vesicle showed that Jag1 is transiently expressed in cells that end up separating the anterior crista from the macula (*Ma and Zhang, 2015*). We found that Lmx1a is one of the factors required for the attenuation of lateral induction: the absence of Lmx1a results in expanded Jag1+ territories in the *Bsd* mouse vestibular system, whilst the overexpression of cLmx1b in the chick inner ear reduces Jag1 expression and results in misplaced sensory patch borders.

Further studies are needed to decipher the interactions between Lmx1a and Notch signalling, and in particular the potential contribution of FGF signalling. In fact several FGF ligands are expressed within and at the boundary of sensory epithelia (*Sánchez-Calderón et al., 2004*; *Olaya-Sánchez et al., 2017*) and the inhibition of FGF signalling can impair sensory patch segregation in zebrafish (*Ma and Zhang, 2015*; *Maier and Whitfield, 2014*). In the chick otic vesicle, FGF inhibition leads to an up-regulation of Jag1 and the Notch effectors Hey1 and Hes5 (*Petrovic et al., 2015*), which might explain the defects in sensory patch segregation observed after FGF blockade in zebrafish. Furthermore, normal inner ear morphology and positioning of the utricle and cristae are severely disrupted in FGF10 null mutants (*Domínguez-Frutos et al., 2011*; *Kopecky et al., 2011*; *Vendrell et al., 2015*). Additional transcription factors such as Otx1/2 (47–50), FoxG1 (*Hwang et al., 2009*; *Pauley et al., 2006*), N-Myc (*Domínguez-Frutos et al., 2011*; *Kopecky et al., 2011*) and Hmx1/3 (*Wang et al., 2004*) are also expressed in non-sensory domains and are required for normal sensory patch segregation. These transcription factors as well as components of the FGF pathway may therefore belong to a complex genetic network that interacts with Lmx1a and Notch signalling to allow sensory patch segregation. Besides their effects on cell differentiation, it would be important to elucidate how these signals impact on cell death or proliferation and the overall morphogenesis of the inner ear – in particular the formation of the tissue constrictions that separate adult sensory organs. In fact, the vestibular system of Lmx1a mutant mice lacks those constrictions and we noticed that the cells overexpressing cLmx1b could form dense aggregates with constricted apical surfaces. This suggests that besides its influence on lateral induction, Lmx1a could have a central role in initiating the complex morphogenetic program leading to the formation of separate chambers for distinct sensory organs.

## Sensory organ progenitors are labile: implications for the mechanisms of sensory organ formation

One influential theory for the formation of inner ear sensory organs is known as the 'compartment boundary' model (*Brigande et al., 2000a*; *Fekete, 1996*). It proposes that long-range patterning signals and cellular interactions could establish several lineage-restricted compartments in the otocyst, which would then develop into distinct structures of the adult inner ear. The boundaries of

these compartments could in turn act as signalling centres, leading perhaps to the formation of new lineage-restricted compartments, giving rise for example to distinct sensory organs (*Brigande et al., 2000a*). This model was first supported by gene expression studies that revealed localized expression of transcription factors, potentially acting as 'selectors' capable of imposing a given identity to a given otic domain. Absence of some of these early regionalised genes causes loss of entire inner ear structures (*Morsli et al., 1999*; *Hammond and Whitfield, 2006*; *Brigande et al., 2000a*), consistent with the notion that inner ear development is partly modular. Fate mapping experiments have also shown that cell mixing is restricted within certain regions of the chick otic placode, implying the existence of clonally-restricted compartments (*Brigande et al., 2000b*; *Sánchez-Guardado et al., 2014*). We also found that some boundaries of Lmx1a-GFP expression are very sharp in the developing otocyst, suggesting limited cell mixing between compartments of distinct identities. However other fate mapping and lineage-tracing studies have led to different conclusions (*Kil and Collazo, 2002*), and some of the early pro-sensory markers (as we also found in the present study) exhibit dynamic changes in their expression pattern (*Cole et al., 2000*; *Neves et al., 2011*; *Sánchez-Guardado et al., 2013*; *Sánchez-Calderón et al., 2004*). Furthermore, lineage tracing experiments in the mouse inner ear have shown that the derivatives of the neuro-sensory otic progenitors can be of different types. The Neurog1-expressing cells of the otic neurogenic domain, for example, give rise to neurons, sensory progenitors of the two maculae, but also to non-sensory cells (*Raft et al., 2007*). Likewise, the progeny of the early (E8.5 to E10.5 in mouse) Sox2-expressing cells can contribute to various non-sensory structures, including the roof of the sensory organs and the semi-circular canals (*Gu et al., 2016*). In the present study, a similar mixture of sensory and non-sensory derivatives was observed among the progeny of the early Notch-active cells in the developing chick inner ear. In summary, it is possible that some strict boundaries do form at an early stage of otic development, but the early sensory-competent domains, at least as defined by Sox2 expression and Notch activity, are not lineage-restricted compartments of cells with a fixed identity. Instead, these domains contain labile, bi-potent cells, which are sensory competent but not necessarily committed (*Slack, 1991*) to the sensory fate. If Notch activity is maintained in these cells, they will eventually become sensory progenitors, as seen in the gain-of-function experiments demonstrating ectopic sensory patch formation after Notch induction at early developmental stages (*Pan et al., 2010*; *Daudet and Lewis, 2005*; *Hartman et al., 2010*; *Liu et al., 2012*). However if Notch is downregulated, or if Lmx1a/cLmx1b expression is maintained or upregulated, they will be diverted towards a non-sensory fate and lead to sensory organ segregation (*Figure 11*). An important question for future investigations is that of the exact transcriptional and epigenetic modifications that underlie the transition from sensory specification (i.e. the potential to become sensory) to the irreversible commitment to the prosensory fate.

## Sensory patch boundary formation is a dynamic and regulated process

When do lineage-restricted boundaries form at the edge of sensory organs? We did not perform the time-controlled lineage tracing experiments that would be required to answer these questions unambiguously, but several observations suggest that this process occurs progressively and is not synchronous throughout the inner ear. The GFP is a relatively stable protein and its fluorescence (in the absence of antibody immunostaining) provides an indication about the present, as well as past expression of *Lmx1a* in the *Lmx1a*$^{GFP/+}$ mice. In the organ of Corti, the medial boundary of GFP expression is very sharp, with cells exhibiting either high or low levels of fluorescence, suggesting the position and identity of these cells has been fixed for some time. In contrast, the lateral domain shows a smooth gradient of decreasing GFP fluorescence overlapping with Sox2 expression. This suggests that the sensory progenitors of the lateral domain could derive from Lmx1a-expressing cells that gradually, albeit in a relatively coordinated manner, adopt a sensory progenitor fate. At the border of the utricle facing the cristae, the so-called 'transitional cells' that express varying levels of GFP are intermingled with GFP- cells at late embryonic stages and up to at least P5, suggesting a relatively slow and asynchronous loss of Lmx1a expression in this domain. Some of the GFP+ cells in this mosaic domain also exhibit Sox2 expression, suggesting that they could represent cells at different stages of commitment towards a sensory fate. This is consistent with the results of a recent single-cell RNA-Seq study in the neonate mouse utricle, which identified a population of cells, presumably corresponding to the transitional cells, with a mixed sensory and non-sensory gene expression profile (*Burns et al., 2015*). The transitional cells of the neonatal (but not the adult)

utricle are also capable of differentiating into hair cells after Atoh1 induction (*Gao et al., 2016*), indicating that these cells are not fully committed to a non-sensory cell fate at this stage. Our results suggest that the hair cells produced at the edge of the neonate mouse utricle (*Burns et al., 2012*) are not derived from early-specified and Sox2-expressing prosensory cells, but perhaps from cells of the Lmx1a-lineage that convert into sensory progenitors at later stages of development. Notch signalling could underlie this developmental switch: Notch activity inhibits cLmx1b expression in the chick otocyst (our data and *Abello and Alsina, 2007*), and a similar antagonism might persist at later stages in the utricle, or at the lateral border of the organ of Corti. As hair cells are being added to the sensory epithelia, both lateral induction (Jag1) and lateral inhibition (Dll1) could contribute to the elevation of Notch activity at the border of developing sensory patches. There is also a central to peripheral progression of hair cell differentiation in the cristae (*Slowik and Bermingham-McDonogh, 2016*; *Slowik and Bermingham-McDonogh, 2013*) suggesting that this process might be a common feature in developing inner ear sensory epithelia. The adult sensory organs of fish and amphibians can add new hair cells at their periphery throughout their adult life (*Corwin, 1981*; *Corwin, 1985*), and it would be interesting to know if this is due to a persistence of the developmental processes proposed here: a production of new sensory cells by lateral recruitment of uncommitted progenitors, as opposed to a growth 'from within', consecutive to the proliferation of committed sensory progenitors.

## Tissue boundary formation in the inner ear: old genes, new tricks

The evolutionary conservation of the molecular players controlling mechanosensory cell formation in Drosophila sense organs and the inner ear has long been recognized (*Eddison et al., 2000*; *Adam et al., 1998*; *Jarman and Groves, 2013*; *Pierce et al., 2008*). In the context of sensory patch segregation, some analogies with the fly exist yet again, but this time with the embryonic wing disc – the tissue in which embryonic compartments were first identified (*Garcia-Bellido et al., 1973*). In the wing disc, two lines of demarcation are established sequentially along the antero-porterior and dorso-ventral axis to create lineage-restricted compartments (reviewed in *Blair, 1995*; *Dahmann et al., 2011*). At the dorso-ventral boundary, two players are essential for this process: Notch signalling and Apterous, the fly homologue of Lmx1a. Remarkably, in both the inner ear and the wing disc, Notch functions by lateral induction to position tissue boundaries. In the wing disc, the dorsal and ventral compartments express a different Notch ligand, respectively Serrate or Delta. However, Notch activity is elevated at the boundary (*de Celis et al., 1996*) due to the activity of Fringe, which potentiates Delta/Notch signalling by glycosylation of the Notch receptor (*Panin et al., 1997*; *Fleming et al., 1997*). In the inner ear, there is also differential Notch activity at the border of the sensory patches, but this is presumably due to the absence of Notch ligand expression in the non-sensory cells. The potential implication of Fringe proteins, which are expressed in the prosensory domains (*Cole et al., 2000*; *Adam et al., 1998*) and at the lateral border of the vestibular patches (*Burns et al., 2015*), remains unexplored in this context. In the wing disc, Apterous acts as a selector gene for the dorsal fate and in its absence, the prospective dorsal cells adopt a ventral fate (*Diaz-Benjumea and Cohen, 1993*). In the ear, Lmx1a may have a similar selector function: in its absence, some, but not all cells of the Lmx1a lineage develop as sensory cells; conversely, overexpression of cLmx1b can divert prospective sensory cells towards non-sensory fate. However, this effect was not consistent, suggesting the implication of additional factors in the commitment to a non-sensory fate. Another major difference lies in the fact that Lmx1a is itself subject to regulation by Notch in the inner ear: its expression does not necessarily predict a non-sensory cell fate, in particular at the borders of developing sensory patches, where Lmx1a is gradually down-regulated as sensory patches expand. In other words, Lmx1a does not define an early lineage-restricted compartment of cells committed to a non-sensory fate.

The analogies between patterning of the fly wing disc and the inner ear suggests an example of 'deep homology': the development of these morphologically divergent structures is regulated by similar molecular players (*Carroll, 2008*), belonging to an ancient genetic module that directs the formation of tissue boundaries. Yet, our data indicate that the inner ear sensory domains do not derive from lineage-restricted compartments specified at an early stage. Instead, their development is a regulated and gradual process, involving dynamic changes in gene expression and cell character.

## Conclusion

The mechanisms that shape the inner ear sensory organs operate on otic progenitor cells that can remain uncommitted for a relatively long period of time, instructing them to switch their genetic program and differentiate along either a sensory or non-sensory lineage. Given the diversity of sensory organ configurations across and within different classes of vertebrates (*Schulz-Mirbach and Ladich, 2016*; *Smotherman and Narins, 2004*; *Retzius, 1881*, *1884*), one could imagine a scenario in which their evolutionary variations in size, number, position and shape might have been facilitated by the developmental lability of their otic progenitors. Gene mutations, duplications, or modifications of cis-regulatory elements targeting the molecular factors controlling the growth and developmental segregation of sensory organs could have contributed to the emergence of new sensory patches and their functional diversification in the course of evolution. Comparative genomic studies focused on the genes essential for prosensory specification and segregation could provide some insights into this question. Further work should also investigate the molecular signals that regulate cell differentiation and proliferation at the border of sensory patches, which could be very relevant to the fields of inner ear stem cell biology and hair cell regeneration.

# Materials and methods

## Transgenic animals and breeding

$Lmx1a^{GFP/GFP}$ mice (*Griesel et al., 2011*) were kindly provided by Dr Ahmed Mansouri (Max Planck Institute for Biophysical Chemistry, Gottingen, Germany). Mice were maintained at the UCL Ear Institute animal facility as heterozygotes on a C57BL/6 mixed background and were bred to generate $Lmx1a^{GFP/GFP}$ (homozygous/Lmx1a null), $Lmx1a^{GFP/+}$ (heterozygous) and $Lmx1a^{+/+}$ (wild type) littermate controls. Animals were genotyped as described in (*Griesel et al., 2011*). The Belly-Spots and Deafness (*bsd*) Lmx1a mutant nice were maintained at Kings College London and bred and genotyped as described in (*Steffes et al., 2012*). The Bsd allele is a genomic deletion encompassing the entire exon3, predicted to result in a frameshift in the translation of exon four and the production of a truncated (91 amino-acids long) Lmx1a protein lacking the LIM2 domain and homeodomain.

All mice were bred and sacrificed using schedule one techniques at various ages between E13 and P40 in accord with United Kingdom legislation outlined in the Animals (Scientific Procedures) Act 1986.

## Plasmids

The plasmid driving transient expression of the Tol2 Transposase, the empty pT2K-CAGGS, and the pT2K-eGFP Tol2 plasmid were previously described (*Sato et al., 2007*). The Tol2-Hes5:bicolor construct was generated by inserting a H2B-mCherry-2A coding sequence downstream of the mouse Hes5 promoter and upstream of a nuclear-localized, destabilized form of eGFP (nd2eGFP) in a pre-existing Tol2-Hes5::nd2eGFP plasmid (*Chrysostomou et al., 2012*). To do so, we first cloned into TOPO (Invitrogen, Thermo Fisher Scientific, UK) a PCR-amplified H2B-mCherry-2A PCR fragment with primers containing BglII sites (underlined): forward CAC<u>AGATCT</u>GAGCCACCATG and reverse T<u>AGATCT</u>AGGTCCTGGGTTCTC. After verifying the sequence of the insert, the BglII-digested fragment was ligated into the BamHI-digested Tol2-Hes5::nd2eGFP plasmid.

The Tol2-red-to-green plasmid was generated by inserting into the EcoRI site of the pT2K-CAGGS Tol2 plasmid a 3 kb EcoRI-digested DNA insert containing the DSRed(floxed)ntreGFP cassette, isolated from the original pTol-EF1-DSred(floxed)ntreGFP (*Hans et al., 2009*). The Hes5-Cre plasmid was generated by inserting the mouse Hes5 promoter sequence (SacI-digested) into the SacI site of a TOPO vector upstream of a tdTomato-IRES-Cre recombinase cassette, which was itself amplified and cloned into TOPO using the high-fidelity *pfu* DNA polymerase (Promega, UK), using as a template a previously described CMV-tdTomato-IRES-Cre plasmid (*Robel et al., 2011*). The pT2K-cLmx1b-IRES-eGFP plasmid was generated by inserting the full-length coding sequence of the chicken Lmx1b gene (digested out from a pBSK-cLmx1b plasmid, a kind gift of Berta Alsina) upstream of an IRES-eGFP coding sequence and downstream of the CAGGS promoter of a Tol2 plasmid. The plasmid driving constitutive expression of the chicken Notch1 Intracellular Domain (NICD) and mRFP1 was previously described in (*Chrysostomou et al., 2012*). The identity of all

plasmids was confirmed by DNA sequencing. All experiments were performed with control plasmids used at the same concentration as the experimental ones.

All plasmids used for in ovo electroporation were purified using the Pureyield Plasmid Midiprep System (Promega, UK), and further concentrated using ethanol-acetate precipitation.

## Chick embryology and in ovo electroporation

Fertilised White Leghorn chicken (*Gallus gallus*) eggs were obtained from Henry Stewart UK and incubated at 38°C for the designated times. Embryonic stages are either from Hamburger-Hamilton (HH) tables (*Hamburger and Hamilton, 1951*) or refer to embryonic days (E), with E1 corresponding to 24 hr of incubation. Embryos older than E5 were sacrificed by decapitation. All procedures were approved by University College London and by the UK Home Office.

The *in ovo* electroporation of the chicken otic cup was performed at E2 (stage HH 12–14) as previously described (*Chrysostomou et al., 2012*; *Freeman et al., 2012*; *Chrysostomou et al., 2012*).

The total concentration of plasmid DNA was kept below 3 mg/ml, and ranged for each individual plasmid between 0.5–1 mg/ml. For the lineage tracing experiments, plasmid DNA concentrations (in mg/ml) were as follows: Hes5::Cre = 0.5; Tol2 transposase = 0.8; Tol2-red-to-green=0.5. Unless otherwise specified, a minimum number of 6 successfully transfected samples were examined for each experimental condition.

## Immunohistochemistry and In situ hybridisation

Entire embryos or inner ear tissue were collected at various developmental stages, fixed for 20 min to one hour in in 0.1 M phosphate buffered saline (PBS) containing 4% paraformaldehyde (PFA), and processed either for whole-mounts preparation or cryosectioning.

For whole-mount preparations, fixed samples were washed three times in PBS and then dissected to expose the otic epithelium before immunostaining. For chicken embryos aged E3-E5, the embryo was bisected along the midline, the hindbrain was removed, and the region surrounding the otocyst was only partially trimmed to facilitate orientation. A small opening was made at the dorsal tip of the otocyst using a fine needle and the tissue was permeabilized in PBS containing 0.3% Triton and 10% goat serum for 30 min before immunostaining using standard procedures (*Daudet and Lewis, 2005*). After immunostaining, Lmx1-GFP or Sox2 immunofluorescence was used to visualize (under a Leica MZ16F fluorescence stereomicroscope), finely dissect and orient the otic tissue during mounting in Vectashield. A fine layer of vacuum grease was applied between the slide and coverslip to avoid excessive flattening of the tissue. For cryosectioning, the tissue was fixed for 30 min in 4% PFA at room temperature and subsequently washed over 30 min in before cryoprotection with sucrose (10%–30% solutions diluted in 0.1 M PBS + 0.02% tween 20). The tissue was then embedded in OCT (Tissue-Tek), frozen by immersion in cold (−70C) isopentane, then stored at −80C until further use. Frozen tissue sections were cut at a thickness of 12–15 µm. Specimens were visualized and imaged using a Zeiss LSM510 inverted confocal microscope or a Perkin Elmer spinning disc confocal.

The following antibodies were used: rabbit anti-Jagged 1 (Santa-Cruz Biotechnology, Dallas, TX; sc-8303; 1:50-1:100), rabbit anti-Sox2 (Abcam, UK; 97959, 1:500), goat anti-Sox2 (Santa-Cruz Biotechnology, Dallas, TX; sc-17320; 1:500), mouse IgG1 monoclonal anti-Sox2 (BD Biosciences, San Jose, CA;561469, 1:500) monoclonal mouse IgG1 anti-HCA (hair cell antigen; a kind gift of Guy Richardson, used at 1:1000-1:2000), monoclonal mouse anti-Myo7a (Developmental Studies Hybridoma Bank; 1:250). Fluorescent Alexa-conjugated Phalloidin (10 µM) and secondary goat antibodies (1:1000) were obtained from Thermo Fischer Scientific (UK).

For *cLmx1b* in situ hybridisation (ISH), the pBSK-cLmx1b plasmid containing the full-length coding sequence of the chicken *Lmx1b* gene (*Abelló et al., 2010*) was used as a template to generate DIG-labelled riboprobes, as previously described (*Daudet and Lewis, 2005*). A minimum of three biological replicates was analysed for each experimental condition and stage.

## Quantification of sensory patch segregation

Chicken embryos were collected at specified stages and their otocysts immunostained for Sox2 expression as described previously. Confocal stacks were collected from whole-mount preparations and z maximal intensity projections were generated and analysed using the Fiji software. The surface area of the segregating cristae and the longest distance separating them from the large prosensory

domain were measured by tracing manually the outline of the cristae and a line between the border of the cristae and the edge of the prosensory domain. The numerical values were exported to Excel and box plots were generated using OriginPro 2017.

## Acknowledgements

We thank Beatriz Lorente-Canovás and Karen Steel for generously providing us with inner ear tissue from the *bsd* mutant mice, and Guy Richardson (hair cell antigen antibody), Berta Alsina (cLmx1b probe), Ahmed Mansouri (Lmx1a-GFP mouse), Michael Brand and Stefan Hans (red-to-green plasmid), Magdalena Götz (CMV-tdTomato-IRES-Cre plasmid), Yoshiko Takahashi and Koichi Kawakami (originalTol2 plasmids) for sharing reagents with us. This work was supported by the Biotechnology and Biological Sciences Research Council (BB/L003163/1 to ND/ZM), the Erasmus programme (HG and DP) and Action on Hearing Loss (summer studentship to EC and International Research Grant G76 to MŻ). We thank Miriam Gomez for excellent technical support. The Myo7A 138-A antibody developed by DJ Orten (Boys Town National Research Hospital) was obtained from the Developmental Studies Hybridoma Bank, created by the NICHD of the NIH and maintained at The University of Iowa, Department of Biology, Iowa City, IA 52242. We thank Berta Alsina for helpful discussions and support at the onset of this project, and Andy Forge for comments on the manuscript.

## Additional information

### Funding

| Funder | Grant reference number | Author |
| --- | --- | --- |
| Biotechnology and Biological Sciences Research Council | BB/L003163/1 Project grant | Zoe F Mann<br>Nicolas Daudet |
| Action on Hearing Loss | G76 Postdoc grant | Nicolas Daudet |
| Erasmus Programme | | Héctor Gálvez<br>David Pedreno |

The funders had no role in study design, data collection and interpretation, or the decision to submit the work for publication.

### Author contributions

Zoe F Mann, Conceptualization, Resources, Formal analysis, Investigation, Visualization, Methodology, Writing—original draft, Writing—review and editing; Héctor Gálvez, David Pedreno, Formal analysis, Investigation, Methodology; Ziqi Chen, Elena Chrysostomou, Formal analysis, Investigation, Visualization, Methodology; Magdalena Żak, Formal analysis, Validation, Investigation; Miso Kang, Formal analysis, Validation, Investigation, Visualization; Elachumee Canden, Resources, Validation, Investigation; Nicolas Daudet, Conceptualization, Resources, Formal analysis, Supervision, Funding acquisition, Investigation, Visualization, Methodology, Writing—original draft, Project administration, Writing—review and editing

### Author ORCIDs

Nicolas Daudet http://orcid.org/0000-0002-4039-4716

### Ethics

Animal experimentation: All experimental procedures were carried out in accordance with the United Kingdom Scientific Procedures Act of 1986. All animals were handled according to protocols covered by a Home Office Animal Procedures Licence (PPL 70/8144) and approved by University College London local Ethics Committee.

### Decision letter and Author response

Decision letter https://doi.org/10.7554/eLife.33323.022
Author response https://doi.org/10.7554/eLife.33323.023

## Additional files

**Supplementary files**
• Transparent reporting form
DOI: https://doi.org/10.7554/eLife.33323.020

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
