## [Decision Letter]

[Editors’ note: a previous version of this study was rejected after peer review, but the authors submitted for reconsideration. The first decision letter after peer review is shown below.]

Thank you for choosing to send your work, "Shaping of inner ear sensory organs by antagonistic interactions between Notch signalling and Lmx1a", for consideration at *eLife*. Your initial submission has been assessed by a Senior Editor in consultation with a member of the Board of Reviewing Editors. Although the work is of interest, we regret to inform you that the findings at this stage are too preliminary for further consideration at *eLife*.

Specifically, the reviewers thought that the paper had many interesting aspects but wanted to see a deeper characterization of the roles of Notch and Lmx1a given that both had established roles in in ear development. In particular, the reviewers requested a careful time course of *Sox2*, Jag1 and Lmx1 from early stages (invaginating otocyst to the appearance of individual sensory patches) be added to the paper. This would likely suggest roles in refinement and maintenance of these domains rather than exclusive involvement in induction. Although we cannot consider the present manuscript further, we would encourage you to submit a new manuscript that addresses the comments of all three reviewers below. If you feel you can address their concerns, we would make every effort to return the manuscript to the same reviewers.

Reviewer #1:

This manuscript proposes that mutual antagonism between the Lmx1 transcription factor and Notch signaling allows the segregation of sensory regions of the amniote inner ear into distinct territories, and that when Lmx1 is reduced, sensory organs either fuse or develop imprecise boundaries.

The main issue with the paper is that although it demonstrates a direct involvement of Lmx1 activity in attenuating Notch signaling (and vice versa), and does so using new methods of visualizing Lmx1 and Notch activity, the main conclusions of the paper are not an enormous advance over several previously published studies of Lmx1 mutant mice.

My main concern with their interpretation is the fact that Jag1, *Sox2* and Notch target genes are expressed broadly in the early otic placode and otocyst, and although these genes later localize to prosensory regions of the ear, they initially mark future sensory and non-sensory domains. Although Notch activity can indeed induce Jag1, it is likely that, under normal circumstances, Notch signaling is preserving prosensory fate when other signals that regulate Lmx1 and other genes become active and down-regulate these genes from broad areas of the ear.

Reviewer #2:

This manuscript addresses an active area of inquiry regarding sensory organ specification in the development of the vertebrate inner ear. Through a series of gain- and loss-of-function analyses in chicken or mouse inner ears, the authors conclude that mutually-inhibitory interactions between a Notch-mediated signaling cascade and the transcription factor, Lmx1b, will cause a common prosensory epithelium to split into separate organs. Cells retaining higher levels of Jag1 become sensory, and those retaining the highest levels of Lmx1b become non-sensory. Disruption of this network of mutual inhibition leads to disruptions at the borders of the sensory patches, formation of ectopic sensory patches, and in some cases foci of non-sensory patches within the sensory organs. The authors discuss their work in the context of two theoretical models of sensory organ specification that have been put forward: a "boundary" model and "sensory segregation" model. The discussion of how their work can inform these theoretical frameworks makes for an intellectually satisfying context for considering these new experimental data. The paper is supported by numerous experiments which have required the authors to generate new constructs, as well as further analysis of existing Lmx1 mouse mutants. This is an important paper that offers researchers new ways of thinking about the mechanisms involved in generating multiple distinct sensory organs in the inner ear during development and evolution. Specifically, it now connects Notch signaling and Lmx1, which had each individually been linked with sensory organ separation in either zebrafish or mouse models.

Major comments:

1) Throughout the manuscript, the authors often refer to "lateral borders" of sensory organs. It is not always clear if this refers to a lateral direction relative to the midline of the body, or if this is simply a way to describe any edge of any organ (excepting the cochlea, where the directionality is clear). Subsection “A gain of lateral induction leads to defects in the positioning of sensory patch boundaries and to sensory domain fusion.” states.…"groups of ectopic hair cells…were found at the lateral border of sensory organs". Subsection “Sensory patch boundary formation is a dynamic and regulated process” discusses sharpening of the edge of the utricle at its "lateral border". Is this the only edge for which this sharpening takes place? The reason this is of interest is that the boundary model offered a hypothesis in which sensory organs arise from within the broader sensory-competent region, but at the boundary with an adjacent compartment. But each sensory organ is not itself considered to be a lineage compartment. Rather, the edge of the organ that abuts an adjacent compartment is expected to be the true lineage boundary. As such, the other edges of the same organ need not be lineage-restricted, and thus could be dependent on the sort of Notch-mediated lateral induction described here to allow for enlargement over time. Moreover, the sensory segregation model leaves open the question of whether the boundary of the entire neurosensory domain might be a lineage boundary, even if it includes future non-sensory cells and neuroblasts that will emigrate away.

2) Can the authors state with any certainty, based on their available data, if EVERY edge of EVERY organ is labile and requires the expression of Lmx1 in its adjacent non-sensory cells to become sharp? The Lmx1a expression data of Figure 8 do not show the gene completely surrounding each organ, but the timing of its expression might be a critical variable.

3) Given the working model presented by the authors, it would be valuable to perform a time course showing Lmx1 and Jag1 expressions side-by-side (preferably in double labeling) for either mouse or chicken. This could help to solidify the conclusion that Lmx1 is intimately associated with the separation of organs as they make their appearance.

Reviewer #3:

Differentiating the developing otocyst into a complicated vertebrate ear requires the formation of sensory epithelia separated by non-sensory epithelium. How this process is regulated at a molecular and cellular level has only been partially elucidated by previous work of the senior author and by work of others showing that loss of certain transcription factors causes fusion of some sensory epithelia, notably the loss of Lmx1a. Building on this existing data, the current work demonstrates that a balance of Notch signal regulation may be at the basis for this sensory epithelium segregation. The paper makes very intelligent use of several new reagents such as bicolor fluorescent reporter for Notch activity and others in chicken. These data suggest that some early Notch positive cells become non-sensory cells and further work established that the precise regulation of lateral induction and localized inhibition are essential for sensory patch segregation.

Following the analysis of chicken ear, the authors re-analyze the Lmx1a ear defects using a GFP reporter system for observations not covered fully by the previous work, Specifically, using the GFP reporter of Lmx1a expression they report that expanded *Sox2* positive domains previously reported contain Lmx1a-GFP cells. This supporter previous suggestions that absence of Lmx1a allows non-sensory cells to acquire a different fate.

Consistent with prolonged proliferation of the utricle is the final observation of a lasting transformation along the utricular boundary.

The Discussion section misses some information pertinent to the point they aim for, namely that loss of n-Myc and thus reduction of proliferation can cause fusion of sensory epithelia (Kopeck et al., 2011). The authors may want to add this information into their model.

Overall, the paper is well written and illustrated but requires additional work on relevant observations provided by others and should be put more specific into the context of tetrapod ear evolution. Citations are provided below that should help guide the authors. While the data build on suggestions raised by previous work on the importance of Lmax1a to suppress the neurosensory fate, the current work goes beyond those suggestions and provides for the first molecular cues how that could work.

some additional data presentation of information that likely is available is suggested below.

[Editors’ note: what now follows is the decision letter after the authors submitted for further consideration.]

Thank you for resubmitting your work entitled "Shaping of inner ear sensory organs by antagonistic interactions between Notch signalling and Lmx1a" for further consideration at *eLife*. Your revised article has been favorably evaluated by Marianne Bronner (Senior editor), a Reviewing editor, and two reviewers.

The manuscript has been improved but there are some remaining minor issues that need to be addressed before acceptance. I hope you can make some textual modifications as described below:

1) The manuscript would benefit from some of the broader insights that highlight the importance of the segregation for distinct specification of a given sensory organ to acquire novel functions (duplicate and diversify (Elliott et al., 2017; Chagnaud et al., 2017).

2) The emphasis on the δ/notch signal in combination with Lmax1a is obvious but the authors may also want to consider the jagged and patchy appearance of the organ of Corti in several mutants reducing FGF, Gata3, miRNA and Atoh1 signaling. Given that Atoh1 signaling is one driver of Hes5 it makes sense to perhaps include some of that data (Jahan et al., 2015).

---

## [Author Response]

[Editors’ note: the author responses to the first round of peer review follow.]

Reviewer #1:This manuscript proposes that mutual antagonism between the Lmx1 transcription factor and Notch signaling allows the segregation of sensory regions of the amniote inner ear into distinct territories, and that when Lmx1 is reduced, sensory organs either fuse or develop imprecise boundaries.The main issue with the paper is that although it demonstrates a direct involvement of Lmx1 activity in attenuating Notch signaling (and vice versa), and does so using new methods of visualizing Lmx1 and Notch activity, the main conclusions of the paper are not an enormous advance over several previously published studies of Lmx1 mutant mice.

We believe that this work provides several new and important insights into the mechanisms of early inner ear development. Anatomical studies of inner ear development in fish and amphibians have led to the proposition that sensory organs are formed by segregation from a common pan-sensory domain, but this process has not been conclusively demonstrated in amniotes. Furthermore, the nature of the molecular mechanisms underlying this process are unclear. In this study, using a whole-mount preparation method for visualizing the expression of *Sox2* and Jag1 in the embryonic chicken otocyst, we provide strong evidence that segregation leads to the formation of the anterior and lateral cristae. We show that sensory patch segregation is regulated by the mutually antagonistic relationship between signals promoting (Notch) or antagonizing (Lmx1) commitment to the sensory fate. This study provides a molecular explanation for the sensory patch patterning defects found in the Lmx1a mouse mutant ear, and clarifies the role of Notch activity in promoting commitment of the sensory-competent cells of the early otocyst to a sensory fate. Our findings emphasize the labile character of the early sensory progenitors of the inner ear, in particular at sensory patch borders, which is a conceptual shift from the somehow deterministic view of sensory organ specification that tends to dominate the field.

My main concern with their interpretation is the fact that Jag1, Sox2 and Notch target genes are expressed broadly in the early otic placode and otocyst, and although these genes later localize to prosensory regions of the ear, they initially mark future sensory and non-sensory domains. Although Notch activity can indeed induce Jag1, it is likely that, under normal circumstances, Notch signaling is preserving prosensory fate when other signals that regulate Lmx1 and other genes become active and down-regulate these genes from broad areas of the ear.

We do not disagree with this comment. It is well known that expression of Jag1, Lmx1a, and a number of other genes is very dynamic in the otic placode and early otocyst, but there is, as this referee points out, a progressive stabilization of their spatial pattern of expression. In this study, we focused on those dynamic changes and how they contribute to the regulation of sensory patch segregation and the positioning of their boundaries at relatively late stages of inner ear development. We believe that the duration and significance of these changes were not fully appreciated in previous studies.

We also agree with the notion that Jag1 (lateral induction) functions predominantly to promote sensory commitment, as opposed to sensory specification – this is one of the conclusions of this study. What we show here is that the formation of multiple sensory patches is critically dependent on the down-regulation of lateral induction, which was not previously known. Lmx1a is probably one of the important factors in opposing lateral induction, but not the only one. We have now emphasized these points in the Discussion.

Reviewer #2:This manuscript addresses an active area of inquiry regarding sensory organ specification in the development of the vertebrate inner ear. Through a series of gain- and loss-of-function analyses in chicken or mouse inner ears, the authors conclude that mutually-inhibitory interactions between a Notch-mediated signaling cascade and the transcription factor, Lmx1b, will cause a common prosensory epithelium to split into separate organs. Cells retaining higher levels of Jag1 become sensory, and those retaining the highest levels of Lmx1b become non-sensory. Disruption of this network of mutual inhibition leads to disruptions at the borders of the sensory patches, formation of ectopic sensory patches, and in some cases foci of non-sensory patches within the sensory organs. The authors discuss their work in the context of two theoretical models of sensory organ specification that have been put forward: a "boundary" model and "sensory segregation" model. The discussion of how their work can inform these theoretical frameworks makes for an intellectually satisfying context for considering these new experimental data. The paper is supported by numerous experiments which have required the authors to generate new constructs, as well as further analysis of existing Lmx1 mouse mutants. This is an important paper that offers researchers new ways of thinking about the mechanisms involved in generating multiple distinct sensory organs in the inner ear during development and evolution. Specifically, it now connects Notch signaling and Lmx1, which had each individually been linked with sensory organ separation in either zebrafish or mouse models.Major comments:1) Throughout the manuscript, the authors often refer to "lateral borders" of sensory organs. It is not always clear if this refers to a lateral direction relative to the midline of the body, or if this is simply a way to describe any edge of any organ (excepting the cochlea, where the directionality is clear). Subsection “A gain of lateral induction leads to defects in the positioning of sensory patch boundaries and to sensory domain fusion.” states.…"groups of ectopic hair cells…were found at the lateral border of sensory organs". Subsection “Sensory patch boundary formation is a dynamic and regulated process” discusses sharpening of the edge of the utricle at its "lateral border". Is this the only edge for which this sharpening takes place? The reason this is of interest is that the boundary model offered a hypothesis in which sensory organs arise from within the broader sensory-competent region, but at the boundary with an adjacent compartment. But each sensory organ is not itself considered to be a lineage compartment. Rather, the edge of the organ that abuts an adjacent compartment is expected to be the true lineage boundary. As such, the other edges of the same organ need not be lineage-restricted, and thus could be dependent on the sort of Notch-mediated lateral induction described here to allow for enlargement over time. Moreover, the sensory segregation model leaves open the question of whether the boundary of the entire neurosensory domain might be a lineage boundary, even if it includes future non-sensory cells and neuroblasts that will emigrate away.

We thank this referee for these very insightful comments.

The terminology used for the ‘lateral border’ of the utricle was not clear enough and has been changed throughout the manuscript. There are indeed some region-specific differences in the pattern of Lmx1a-GFP fluorescence at the edge of the developing utricle (and the prosensory domain from which it derives), which we now describe in the text (see also new Figure 8 and Figure 8—figure supplement 1).

This referee is right in pointing out that the compartment-boundary model does not necessarily imply that sensory organs represent themselves lineage-restricted compartments. Explaining in detail the ‘compartment-boundary’ model would unnecessarily complicate the Introduction and we have now decided to address this topic in the Discussion (Sensory organ progenitors are labile: implications for the mechanisms of sensory organ formation). Our results indeed suggest that the prosensory domains could have different types of boundaries (lineage-restricted and not) at their periphery, and this is now summarized as follows in the Discussion section:

“One could therefore combine the two models for sensory organ formation by proposing that the boundary model (regionalised expression of ‘selector’ genes, regulated by long-range and local cell-cell interactions) operates in the context of a labile, ‘pan-sensory’ domain (Figure 11). […] This process would be regulated by regionalised signals (Notch, Lmx1a, and any factor capable of regulating their expression or activity) that commit sensory-competent cells to a prosensory fate, or divert them from this fate to form non-sensory cells in between sensory patches.”

2) Can the authors state with any certainty, based on their available data, if EVERY edge of EVERY organ is labile and requires the expression of Lmx1 in its adjacent non-sensory cells to become sharp? The Lmx1a expression data of Figure 8 do not show the gene completely surrounding each organ, but the timing of its expression might be a critical variable.

Our data show clear differences in the pattern of Lmx1a-GFP expression at different position along the edge of sensory organs (for example on/off transitions versus graded transition at the medial and lateral border of the organ of Corti or in the developing utricle). We now put more emphasis on these differences in the result section and have provided new images that illustrate these differences during the development of the inner ear (Figure 8 and Figure 8—figure supplement 1). Nevertheless, the mosaicism in Lmx1a-GFP expression is more pronounced and long-lasting at the edge of the utricle (in particular in the region facing the anterior and lateral cristae) than in any other organ we examined, suggesting an enhanced degree of plasticity in this domain (Figure 11).

3) Given the working model presented by the authors, it would be valuable to perform a time course showing Lmx1 and Jag1 expressions side-by-side (preferably in double labeling) for either mouse or chicken. This could help to solidify the conclusion that Lmx1 is intimately associated with the separation of organs as they make their appearance.

We have included new data showing the expression of Jag1/*Sox2* and their relationship to Lmx1a during the development of the inner ear (Figure 1, Figure 1—figure supplement 1, Figure 1—figure supplement 2, Figure 8 and Figure 8—figure supplement 1). These data provide strong evidence that the cristae form by segregation, and provide a more detailed analysis of the relationship between Lmx1a and *Sox2*/Jag1 during the early stages of sensory organ development.

Reviewer #3:Differentiating the developing otocyst into a complicated vertebrate ear requires the formation of sensory epithelia separated by non-sensory epithelium. How this process is regulated at a molecular and cellular level has only been partially elucidated by previous work of the senior author and by work of others showing that loss of certain transcription factors causes fusion of some sensory epithelia, notably the loss of Lmx1a. Building on this existing data, the current work demonstrates that a balance of Notch signal regulation may be at the basis for this sensory epithelium segregation. The paper makes very intelligent use of several new reagents such as bicolor fluorescent reporter for Notch activity and others in chicken. These data suggest that some early Notch positive cells become non-sensory cells and further work established that the precise regulation of lateral induction and localized inhibition are essential for sensory patch segregation.Following the analysis of chicken ear, the authors re-analyze the Lmx1a ear defects using a GFP reporter system for observations not covered fully by the previous work, Specifically, using the GFP reporter of Lmx1a expression they report that expanded Sox2 positive domains previously reported contain Lmx1a-GFP cells. This supporter previous suggestions that absence of Lmx1a allows non-sensory cells to acquire a different fate.Consistent with prolonged proliferation of the utricle is the final observation of a lasting transformation along the utricular boundary.The Discussion section misses some information pertinent to the point they aim for, namely that loss of n-Myc and thus reduction of proliferation can cause fusion of sensory epithelia (Kopeck et al., 2011). the authors may want to add this information into their model.Overall, the paper is well written and illustrated but requires additional work on relevant observations provided by others and should be put more specific into the context of tetrapod ear evolution. Citations are provided below that should help guide the authors. While the data build on suggestions raised by previous work on the importance of Lmax1a to suppress the neurosensory fate, the current work goes beyond those suggestions and provides for the first molecular cues how that could work.

We thank this referee for the positive comments and suggestions for changes. We have taken into account the main suggestions as detailed below.

[Editors' note: the author responses to the re-review follow.]

Thank you for resubmitting your work entitled "Shaping of inner ear sensory organs by antagonistic interactions between Notch signalling and Lmx1a" for further consideration at eLife. Your revised article has been favorably evaluated by Marianne Bronner (Senior editor), a Reviewing editor, and two reviewers.The manuscript has been improved but there are some remaining minor issues that need to be addressed before acceptance. I hope you can make some textual modifications as described below:1) The manuscript would benefit from some of the broader insights that highlight the importance of the segregation for distinct specification of a given sensory organ to acquire novel functions (duplicate and diversify (Elliott et al., 2017; Chagnaud et al., 2017).

This work has some implications for our understanding of the mechanisms of inner ear evolution, but these remain somehow hypothetical in the absence of experimental validation (e.g. comparative and functional genomics studies in different classes of vertebrates), so we decided to focus the Discussion on the developmental/molecular findings instead – which are in themselves significant advances. However, the Introduction includes a fair section on the concept of sensory organ multiplication/diversification in the course of inner ear evolution, refers to several excellent reviews focused on inner ear evolution, and we have now modified the text to specifically mention organ duplication as follows:

“In the course of evolution, a gradual increase in the number of sensory patches (up to 9 in some amphibians), possibly due to the duplication and modification of pre-existing structures, has led to the acquisition of new inner ear functionalities – in particular sound detection in terrestrial vertebrates (5,9,10).”

Furthermore, we have modified the text of the concluding remarks as follows:

“Given the diversity of sensory organ configurations across and within different classes of vertebrates (4,80–82), one could imagine a scenario in which their evolutionary variations in size, number, position and shape might have been facilitated by the developmental lability of their otic progenitors. […] Further work should also investigate the molecular signals that regulate cell differentiation and proliferation at the border of sensory patches, which could be very relevant to the fields of inner ear stem cell biology and hair cell regeneration.”

2) The emphasis on the δ/notch signal in combination with Lmax1a is obvious but the authors may also want to consider the jagged and patchy appearance of the organ of Corti in several mutants reducing FGF, Gata3, miRNA and Atoh1 signaling. Given that Atoh1 signaling is one driver of Hes5 it makes sense to perhaps include some of that data (Jahan et al., 2015).

We have referred to other pathways, such as FGF, which have been linked to prosensory specification and the control of sensory patch size/morphology. As far as we are aware, the potential role of Atoh1 in this context has not been demonstrated (including in Jahan et al., 2015) but could be worth exploring.